# Ontogeny of arterial macrophages defines their functions in homeostasis and inflammation

Tobias Weinberger[1,2], Dena Esfandyari [2,3,12], Denise Messerer[1,2,12], Gulce Percin[4,12], Christian Schleifer[1,2,12], Raffael Thaler[1,2,12], Lulu Liu[1,2], Christopher Stremmel [1,2], Vanessa Schneider[1,2], Ronald J. Vagnozzi [5], Jennifer Schwanenkamp[5], Maximilian Fischer[1,2], Katrin Busch[6], Kay Klapproth[6], Hellen Ishikawa-Ankerhold[1,2], Lukas Klösges[1,2], Anna Titova[1,2], Jeffery D. Molkentin [5,7], Yasuhiro Kobayashi[8], Stefan Engelhardt [2,3], Steffen Massberg [1,2], Claudia Waskow[4,9,13], Elisa Gomez Perdiguero [10,13] & Christian Schulz [1,2,11✉]

Arterial macrophages have different developmental origins, but the association of macrophage ontogeny with their phenotypes and functions in adulthood is still unclear. Here, we combine macrophage fate-mapping analysis with single-cell RNA sequencing to establish their cellular identity during homeostasis, and in response to angiotensin-II (AngII)-induced arterial inflammation. Yolk sac erythro-myeloid progenitors (EMP) contribute substantially to adventitial macrophages and give rise to a defined cluster of resident immune cells with homeostatic functions that is stable in adult mice, but declines in numbers during ageing and is not replenished by bone marrow (BM)-derived macrophages. In response to AngII inflammation, increase in adventitial macrophages is driven by recruitment of BM monocytes, while EMP-derived macrophages proliferate locally and provide a distinct transcriptional response that is linked to tissue regeneration. Our findings thus contribute to the understanding of macrophage heterogeneity, and associate macrophage ontogeny with distinct functions in health and disease.

[1] Medizinische Klinik und Poliklinik I, Klinikum der Universität München, Ludwig-Maximilians-Universität, Marchioninistrasse 15, 81377 Munich, Germany. [2] DZHK (German Centre for Cardiovascular Research), partner site Munich Heart Alliance, 80802 Munich, Germany. [3] Institute of Pharmacology and Toxicology, Technische Universität München, Biedersteiner Strasse 29, 80802 Munich, Germany. [4] Regeneration in Hematopoiesis, Leibniz-Institute on Aging - Fritz-Lipmann-Institute (FLI), Beutenbergstrasse 11, 07745 Jena, Germany. [5] Department of Pediatrics, Cincinnati Children's Hospital Medical Center, Cincinnati, OH 45229, USA. [6] Division of Cellular Immunology, German Cancer Research Center (DKFZ), Im Neuenheimer Feld 280, 69120 Heidelberg, Germany. [7] Howard Hughes Medical Institute, Cincinnati Children's Hospital Medical Center, Cincinnati, OH, USA. [8] Institute for Oral Science, Matsumoto Dental University, 1780 Hiro-Oka Gobara Shiojiri, Nagano 390-0781, Japan. [9] Faculty of Biological Sciences, Friedrich-Schiller-University Jena, 07737 Jena, 07745 Jena, Germany. [10] Institut Pasteur, Macrophages and Endothelial cells, Département de Biologie du Développement et Cellules Souches, UMR3738 CNRS, Paris 75015, France. [11] Walter-Brendel-Center for Experimental Medicine, Ludwig Maximilian University, Marchioninistrasse 27, 81377 Munich, Germany. [12]These authors contributed equally: Dena Esfandyari, Denise Messerer, Gulce Percin, Christian Schleifer, Raffael Thaler. [13]These authors jointly supervised this work: Claudia Waskow, Elisa Gomez Perdiguero. ✉email: christian.schulz@med.uni-muenchen.de

Macrophages are the most abundant immune cells in the arterial wall. They play an important role in inflammatory vascular diseases such as atherosclerosis and fibrosis. Macrophages can drive the local inflammatory process through secretion of soluble mediators and various interactions with adjacent immune and nonimmune cells[1]. However, they can also provide anti-inflammatory properties that lead to resolution of inflammation[2]. In addition to latter differences on a functional level, heterogeneity of macrophages has recently been identified also on a molecular level by single-cell RNA transcriptomics[3,4]. Thus, macrophages in arteries vary in their cellular identities and functional properties.

A different level of heterogeneity is provided by the developmental origin of arterial macrophages[5]. The first hematopoietic site giving rise to tissue macrophages is the yolk sac (YS). Here, erythro-myeloid progenitors (EMPs) are generated from embryonic day (E)8.5 onwards, and traffic to the embryonic tissues between E9 and E12[6,7]. Consequently, YS-derived macrophages have been identified in most adult organs[8,9], including the cardiovascular system[10–12]. In the mouse aorta, YS hematopoiesis represents the main source of macrophages during embryonic development[10]. After birth, bone marrow (BM) hematopoiesis adds to the pool of arterial macrophages and BM-derived monocytes are thought to represent the main source of macrophages in postnatal life[10]. The quantitative contribution of EMP-derived macrophages to arterial macrophages at different stages of mouse life has been incompletely understood, which is in part due to the lack of fate-mapping models that quantitatively label macrophages of early embryonic origin. Furthermore, experimental mouse studies of vascular inflammation have focused mainly on the recruitment of BM-derived monocytes[13,14]. However, the response of adventitial macrophages to angiotensin-II (AngII) induced vascular fibrosis might be different between EMP- and BM-derived macrophage populations, which could contribute to strategies differentially targeting arterial immune cells.

Here we show that arterial macrophages derive predominantly from YS EMPs, which give rise to a transcriptionally distinct cluster of macrophages with mainly homeostatic and anti-inflammatory properties in steady state as well as in response to AngII inflammation. In adult mice, these EMP-derived macrophages persist for long periods of time, but decrease in absolute numbers in senescence without compensation by BM-derived progenitors. EMP-derived macrophages express high levels of homeostatic genes such as *Lyve-1*, which links them to physiological functions in arteries. Thus the developmental origin of arterial macrophages is associated with differences in their cellular identity and functions in adult mice.

## Results

**EMPs give rise to long-lived arterial macrophages.** To specifically map the fate of YS EMPs, we pulse-labelled *Csf1r^MerCreMer(MCM)Rosa26^eYFP* embryos with 4-hydroxytamoxifen at E8.5. As expected from previous work[10], we detected eYFP+ F4/80^high macrophages with typical morphology in the aorta of E16.5 embryos (Supplementary Fig. 1A). In a histological analysis of adult *Csf1r^MCMRosa26^eYFP* mice pulse-labelled at E8.5, we readily identified YS-derived macrophages in the aorta. They located in the adventitia but did not inhabit intima or media (Supplementary Fig. 1B). Importantly, YS-derived macrophages remained present in the adventitia in 1-year-old mice (Fig. 1a), which supports recent findings on the longevity of tissue-resident macrophages[15]. eYFP labelling efficiency was comparable after 3 and 6 months and similar to other large macrophage populations, such as liver Kupffer cells (Fig. 1b). The quantitative contribution

of YS hematopoiesis to arterial macrophages has been unclear, since the *Csf1r^MCMRosa26^eYFP* model underestimates the contribution of YS precursors to the pool of tissue macrophages[6]. To address this aspect, we consulted additional lineage-tracing models.

The tyrosine kinase Tie2 is expressed in YS progenitors as well as aorta-gonado-mesonephros (AGM), fetal liver and BM HSCs[16]. We carried out tamoxifen pulse-labelling in *Tie2^MCM-Rosa^eYFP* embryos to induce temporally controlled gene expression in *Tie2+* cells and their progeny, allowing to assess their contribution to arterial macrophages. We applied tamoxifen by oral gavage at either day E7.5 or E10.5 to specifically label *Tie2+* cells that arise during YS hematopoiesis (E7.5) or during later stages of hematopoiesis including AGM and fetal liver (E10.5), and analysed labelling efficiency in arterial macrophages of adult mice[6]. Tamoxifen application at E7.5 resulted in increased labelling of arterial and liver macrophages compared to labelling in *Csf1r^MCMRosa26^eYFP* mice. However, only a low number of eYFP+ macrophages was observed when tamoxifen was administered at E10.5 (Fig. 1c, d; Supplementary Fig. 2A, B). Labelling of HSCs was comparable between both timepoints (Supplementary Fig. 2C, D). Thus, a significant proportion of arterial macrophages arises from *Tie2+Csf1r+* EMPs during YS hematopoiesis.

To further address the contribution of EMPs to the arterial macrophage pool, we carried out lineage tracing in *Rank^CreRosa26^eYFP* mice, which efficiently labels YS-derived tissue-resident macrophages but not BM HSCs and their progeny[17]. Consequently brain microglia, which are solely derive from YS hematopoiesis[8], expressed eYFP whereas blood monocytes were not labelled (Fig. 1e, f, Supplementary Fig. 3B, C). We analysed the aorta between the left subclavian artery and the aortic bifurcation, and quantified both relative (% eYFP labeling) and absolute numbers of macrophages at different timepoints. Arterial macrophages were mostly derived from YS EMPs in the first week of life (Fig. 1e–h, Supplementary Fig. 3A), which is confirmative of previous reports[10]. From birth to adulthood, the absolute number of macrophages in the adventitia increased ~3.7 fold (Fig. 1g). This was driven by an increase of mainly EMP-derived macrophages in adolescence as well as through contribution of BM-derived (eYFP−) macrophages later in adulthood. While this results in a relative decline of eYFP+ macrophages, they remain the dominant macrophage population in mice until at least 45 weeks of age. In the process of ageing, EMP-derived macrophages diminish in numbers. However, their loss in 90-week-old mice is not compensated for by BM-derived (eYFP−) macrophages, whose population remained stable in absolut numbers (Fig. 1g). In summary, EMP-derived macrophages are the predominant macrophage population in healthy adult mice but are diminished in the course of ageing.

**Cellular identity of EMP-derived adventitial macrophages.** Next we sought to address the transcriptional diversity of arterial macrophages taking into account their developmental paths. Therefore, we combined single-cell RNA (scRNA) sequencing with lineage tracing in *Rank^CreRosa26^eYFP* mice. We isolated whole immune cell populations (all CD45-positive live cells) from the adventitia of 16-week-old mice ($n = 3$), and sequenced the transcriptome of single cells (Supplementary Fig. 4A–C).

Following quality controls, we analysed 346 single cells using Seurat-V3 based unsupervised clustering. We identified five distinct clusters of immune cells (Fig. 2a; Supplementary Data 1) including B cells (Cluster 2 defined by *Cd79b*, *Cd19*, *Ms4a1*, and *Ighd*) and a combined cluster of T cells and NK cells (Cluster 1

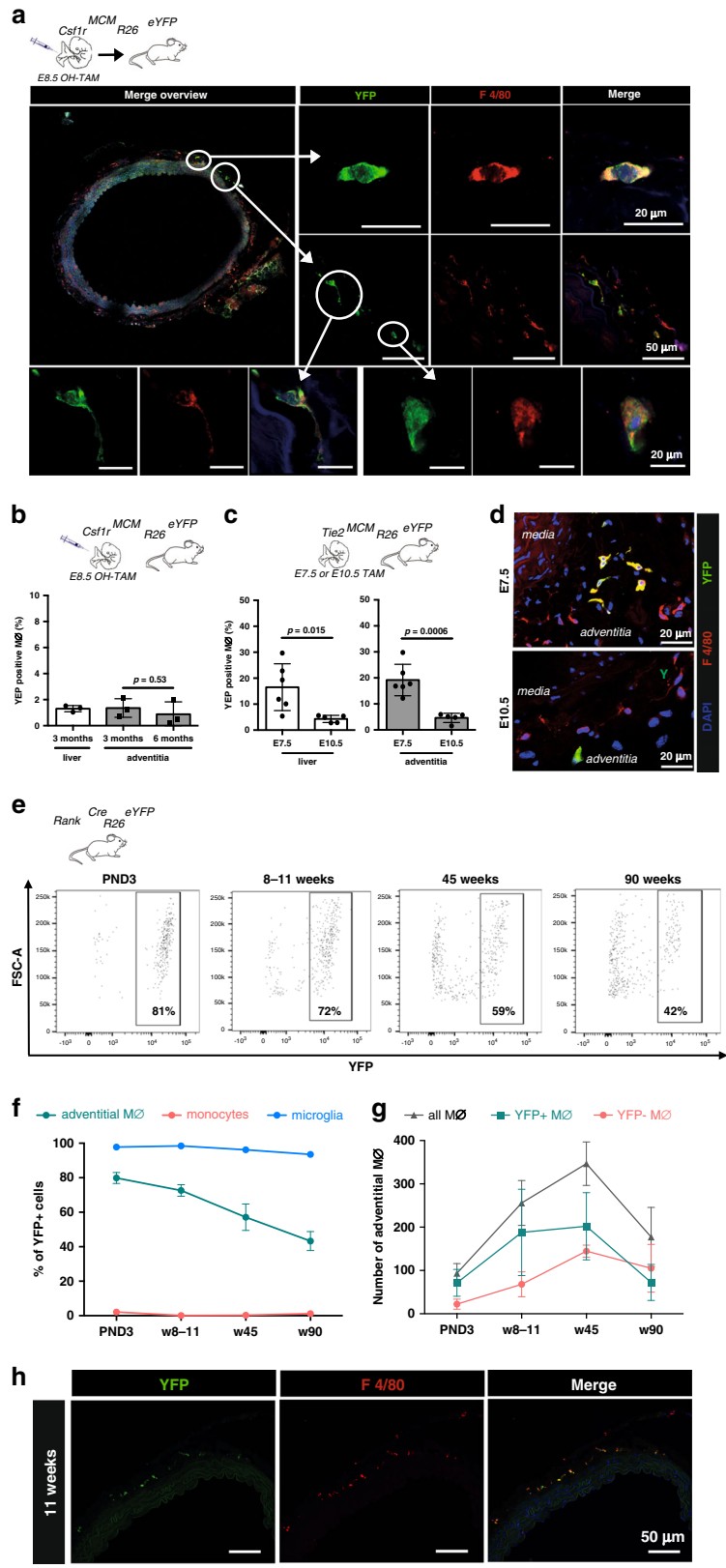

defined by *Trbc2*, *Thy1*, *Cd3g*, Cd3d, and Nkg7) (Fig. 2 and Supplementary Fig. 4F). Macrophages were defined by high RNA expression of *Adgre1* (F4/80) and *Itgam* (CD11b) similar to our flow-cytometry analysis (Supplementary Fig. 4D). They exhibited more heterogeneous gene profiles forming three clusters (Clusters 0, 3, and 4) connected to distinct biological processes (Fig. 2d, e

and Supplementary Fig. 4D, E). By analysing expression of *Yfp*, we confirmed that the majority of macrophages (~55%) derived from EMP-progenitors in adult mice, which is in-line with our initial flow-cytometry analysis (Fig. 2b, c). Further, we mapped EMP-derived macrophages and their contribution to the different clusters (Fig. 2b).

**Fig. 1 Arterial macrophages mainly derive from YS EMPs. a, b** YS EMPs were pulse-labeled by intraperitoneal injection of 4-hydroxytamoxifen (OH-TAM) in pregnant $Csf1r^{MCM}Rosa26^{eYFP}$ mice at E8.5. **a** Image of pulse-labelled eYFP+ macrophages in the adventitia of a 1-year-old mice. **b** Percentage of labelled macrophages in 3- and 6-months-old mice ($n = 3$ at each timepoint; two independent experiments). **c, d** Pulse labelling of early hematopoiesis in pregnant $Tie2^{MCM}Rosa26^{eYFP}$ mice by application of tamoxifen at E7.5 or E10.5. **c** Percentage of labelled cells in 3-month-old mice analysed by flow cytometry (liver) and immunohistology (aorta) is indicated ($n = 6$ for E7.5 and $n = 5$ for E10.5; 2 independent experiments at each timepoint). **d** Representative images of the aorta for each timepoint. **e–g** Flow-cytometry analysis of $Rank^{Cre}Rosa26^{eYFP}$ mice at different timepoints. **e** eYFP expression in macrophages (single CD45+, lin−(CD11c, SiglecF, Ter119, Ly6g), CD11b+, F4/80+ cells) at indicated time-points. **f** Percentage of macrophage eYFP expression in the adventitia, brain microglia, and monocytes from BM (PND3) or blood (PND3 ($n = 3$), weeks (w) 8–11 ($n = 6$), w45 ($n = 3$), w90 ($n = 2$), from 6 independent experiments.). **g** Absolute numbers of eYFP+ and eYFP− macrophages (Mφ) in the adventitia (PND3 ($n = 3$), weeks (w) 8–11 ($n = 6$), w45 ($n = 3$), w90 ($n = 2$), from six independent experiments.). **h** Representative immunofluorescence image (YFP, F4/80) ($n = 3$; 3 inidvidual experiments). Two-sided $t$-test was performed and mean ± SD is shown.

Macrophages in cluster 0 expressed high levels of *Lyve-1*, *Stab1*, and *Gas6*, which have been linked to homeostatic and anti-inflammatory immune functions[10,18–22] (Fig. 2d). Cluster 0 mainly consisted of *Yfp*+ cells, substantiating the notion that homeostatic macrophages in the aorta are mainly derived from YS EMPs. Correspondingly, we observed low expression of *Ccr2*, which characterizes BM-derived inflammatory macrophages (Supplementary Data 1). Further, cluster 0 displayed increased expression of *Pf4*, a marker previously described in resident arterial macrophages (Fig. 2e)[3]. Within cluster 0, the largest macrophage cluster in steady-state conditions, we further compared gene expression between *Yfp*+ and *Yfp*− cells. Interestingly, some genes were differentially expressed between macrophage lineages within this cluster (Supplementary Fig. 5), for example *Lyve-1* expression was increased in EMP-derived macrophages. This supports the notion that ontogeny of macrophages is associated with differences in their cellular identity in adult mice.

Macrophage cluster 4 was exclusively formed by *Yfp*− BM-derived macrophages and defined by high expression of *Ly6c*, *Cxcr4*, *Cx3cr1*, *Nr4a1*, and *Il1ß*. Increased expression of *S100a9* in this cluster may be associated with early stages of differentiation from a monocyte to macrophage phenotype[23]. Cluster 3 was comprised of both *Yfp*− as well as *Yfp*+ macrophages, and showed increased expression of *H2-Ab1*, *H2-Aa*, *Retnla*, *CD74*, and *Ear2*. Furthermore, macrophages of cluster 3 expressed high levels of *Mki67* suggesting a proliferative state of this cluster (Fig. 2d, e). Taken together, the ontogeny of arterial macrophages is associated with distinct transcriptional clustering in single-cell gene expression profiling. While EMP-derived macrophages mainly expressed anti-inflammatory and homeostatic factors, we identified a cluster of early transdifferentiated BM-derived macrophages displaying an inflammatory phenotype (Supplementary Fig. 4E). A cluster of proliferating macrophages was of dual origin from EMPs as well as HSCs and mainly characterized by antigen presentation.

A limitation of our scRNA-seq analysis in steady state is the low number of total cells analyzed. This is due to the low abundance of macrophages in healthy adventitia, which has also been observed in other reports[3,4].

**Contribution of HSCs to arterial macrophages.** Having identified YS EMPs as a major contributor to arterial macrophages in adult mice, we next aimed to confirm our findings taking a complementary approach by quantifying HSC-derived macrophages in the aorta. We analysed expression of the FMS-like tyrosine kinase 3 (*Flt3*) in HSCs and their progeny, which is absent in YS hematopoietic progenitors[6,9]. In adult *Flt3*$^{Cre}$; *Rosa26*$^{eYFP}$ mice ~90% of blood monocytes were labelled, whereas in the aortas of 45-week-old mice ~45% of macrophages were eYFP positive (Fig. 3a, b). Flow-cytometry data were

confirmed by immunohistology (Fig. 3c, d) and support the notion that BM progeny contribute less than half the population of tissue macrophages in the mouse aorta. Notably, eYFP+ and eYFP− macrophages showed an equal distribution in different anatomic regions of the aorta (Supplementary Fig. 6A, B) as well as a random distribution in tissue sections with similar distance to the media and to adjacent macrophages (Supplementary Fig. 7A, B), suggesting that the different populations do not inhabit specified niches.

To determine the proliferation capacity of resident macrophages in the adventitia we injected *Flt3*$^{Cre}$*Rosa26*$^{eYFP}$ mice three times every 48 h with 5-ethynyl-2′-deoxyuridine (EdU), which is incorporated into DNA of proliferating cells. At steady-state conditions, ~5% of arterial macrophages incorporated EdU, which was comparable between eYFP+ and eYFP− macrophages (Fig. 3e, f). In the *Flt3*$^{Cre}$*Rosa26*$^{eYFP}$ mouse model, EdU incorporation in BM-independent (eYFP−) resident macrophages indicates their local proliferation.

To determine the de novo contribution of BM precursors to arterial macrophages in adult mice, we next analysed BM chimeric mice after transplantation with BM HSCs. Replacement of tissues macrophages was previously studied in lethally-irradiated mice, reporting chimerism rates of up to 90% in arterial macrophages[10]. Lethal irradiation is well known to induce tissue inflammation, and also to limit the potential of resident macrophages to self-renew. It is therefore an unsuitable method to determine steady-state turnover of tissue-resident macrophages. We established a genetic BM-transplantation model that harnesses conditional gene deletion to ablate the transcription factor *c-myb* and thereby depletes BM cells, hence allowing BM substitution without other preconditioning and circumventing the critical issues of irradiation mentioned above[24].

YS EMPs develop independently of *c-myb*[9]. Consequently, in E16.5 *c-myb*$^{−/−}$ mice lacking the HSC lineage we readily identified F4/80$^{hi}$ macrophages in the aorta (Supplementary Fig. 8A). In adult *Mx1*$^{Cre}$*c-myb*$^{flox/flox}$;*CD45.2* mice conditional activation of the Mx1 promoter is achieved by application of poly (I:C) resulting in the depletion of *c-myb*-dependent BM precursors[24]. Transplantation of *CD45.1* BM resulted in high chimerism in blood monocytes (Supplementary Fig. 8B). After 3 and 9 months of BM-chimerism, ~20% of macrophages in the aorta expressed CD45.1 (Fig. 3g, h). Thus, we identified a low turnover of arterial macrophages in steady state with replacement of resident macrophages by BM-derived precursors.

Next, we induced labelling in Kit+ progenitors by 4-week tamoxifen feeding of adult *c-kit*$^{MCM}$*Rosa26*$^{GFP}$ mice. After this time, 45% of blood monocytes were labelled and ~5% of arterial macrophages expressed GFP (Fig. 3i, j). In summary, our data indicates, that there is continuous contribution of BM precursors to macrophages in the mouse aorta, as well as local proliferation of tissue-resident macrophages.

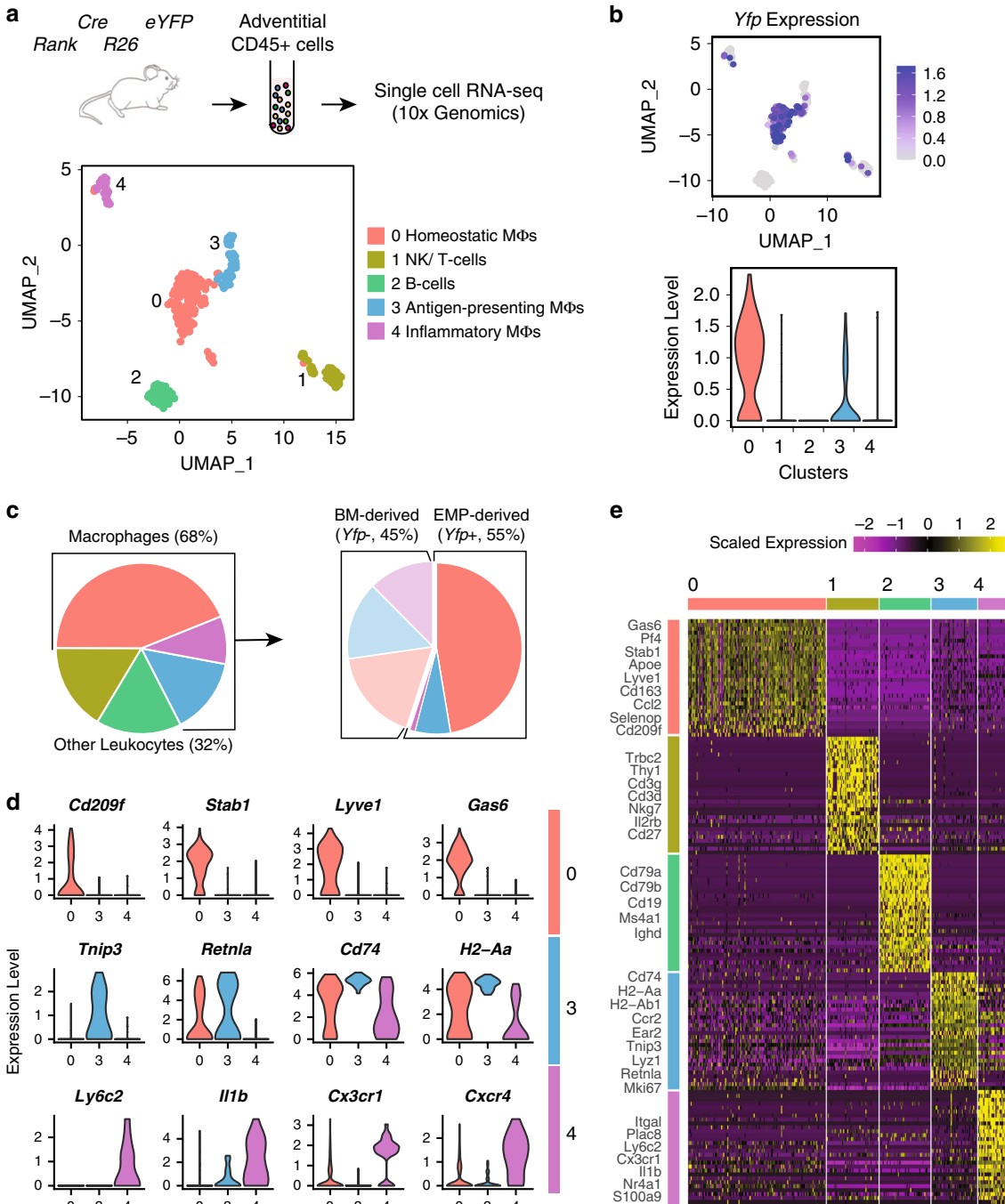

**Fig. 2 Transcriptional profiling of EMP-derived macrophages.** Single-cell RNA sequencing of CD45+ immune cell populations in the adventitia at steady state. **a** Experimental design and UMAP representation of gene expression in single CD45+ cells collected from aortas of $Rank^{Cre}Rosa26^{eYFP}$ mice ($n = 346$ cells from three 16-week-old mice). Cells clustered into five distinct populations, namely different macrophage subtypes, B and T lymphocytes and NK cells. **b** Expression of $Yfp$ in adventitial leukocytes depicted as Feature (upper panel) and Violin (lower panel) plots. Expression of transgene is mainly identified in homeostatic macrophages (cluster 0). **c** Pie charts depicting the proportion of various leukocyte populations in adventitia (left panel) and fraction of $Yfp$-expressing cells within adventitial macrophages (right panel). Macrophages comprise the majority of adventitial leukocytes at steady-state (68% of total). **d** Violin plots representing expression of specific markers of homeostatic (cluster 0), antigen-presenting (cluster 3), and inflammatory (cluster 4) macrophages. **e** Heatmap showing top30 enriched genes in each leukocyte cluster. Expression scale is Log2 fold-change of gene expression in the corresponding cluster compared to all other clusters. Established markers of each cell type are labeled.

**Proliferation of arterial macrophages following depletion.** After having identified differences in the molecular programs of EMP- and BM-derived arterial macrophages, we aimed to explore their response to homeostatic and inflammatory cues. Survival of resident arterial macrophages is dependent on CSF1R as shown in a genetic model targeting $Lyve-1$-expressing macrophages as well as using the pharmacological CSF1R-inhibitor Ki20227[25]. Here, we combined CSF1R-inhibition using PLX5622, which efficiently depletes tissue macrophages[26], with lineage tracing of BM-derived macrophages. Inhibitor feeding resulted in robust depletion of arterial macrophages as expected (Fig. 4a, b). After 3 days of PLX5622-diet, ~75% of tissue macrophages were

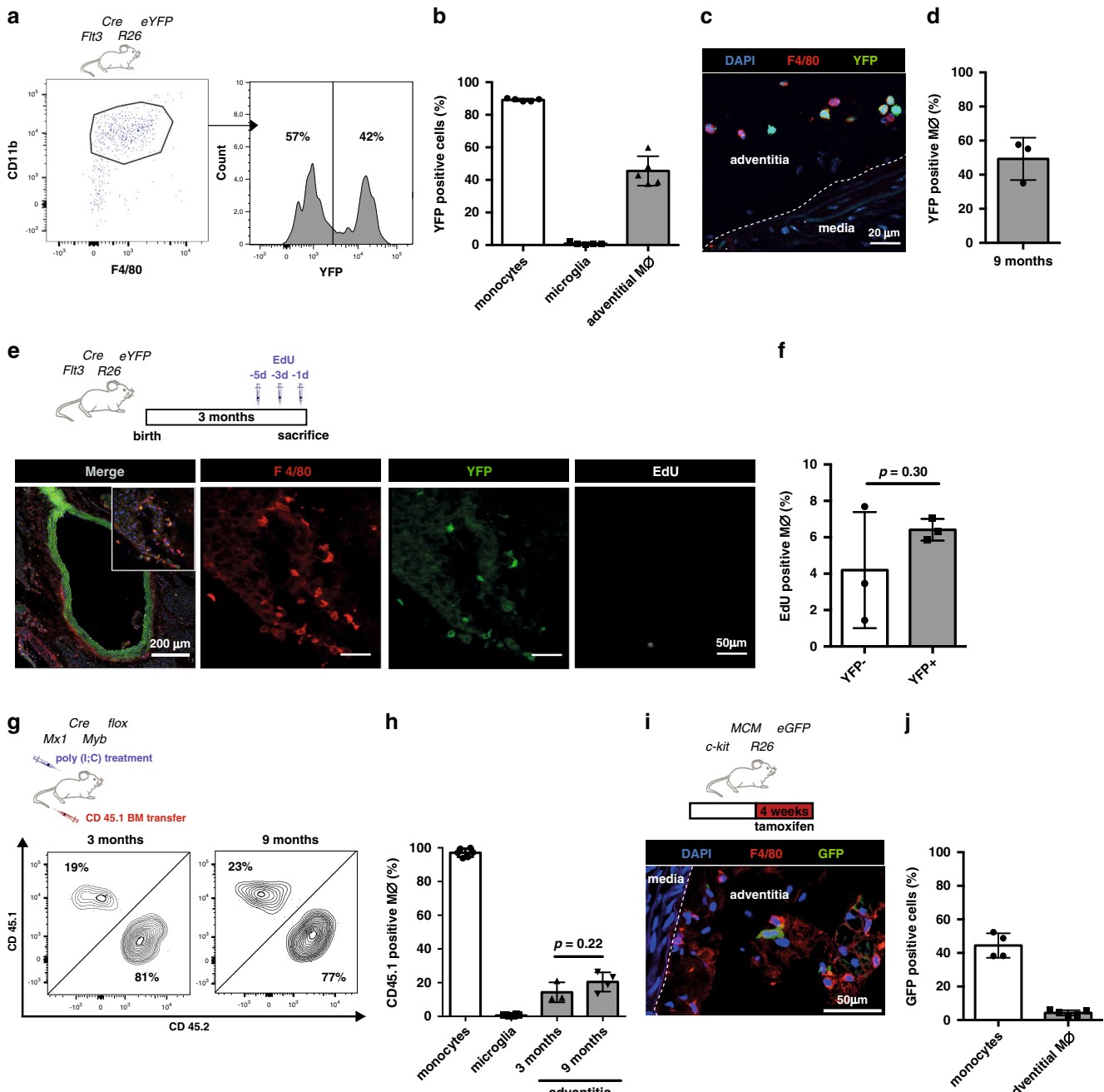

**Fig. 3 HSC-derived arterial macrophages. a, b** Flow-cytometry analysis of arterial macrophages of $Flt3^{Cre}Rosa26^{eYFP}$ mice (gated on single CD45+, lin−(CD11c, SiglecF, Ter119, Ly6G) cells) (left) and characterization of eYFP expression (right). **a** Percentage of eYFP+ monocytes in blood and adventitial macrophages of 45-week-old mice ($n = 5$; two individual experiments). **c** Representative immunohistological images and **d** quantification of eYFP+ macrophages stained for F4/80 ($n = 3$; three individual experiments). **e, f** EdU incorporation in adventitial macrophages of $Flt3^{Cre}Rosa26^{eYFP}$ mice (14–16 week old). **e** Representative immunohistological images and **f** Quantification of EdU incorporation in eYFP− and eYFP+ F4/80+ macrophages ($n = 3$; 3 individual experiments). **g, h** Generation of BM chimeric mice by injection of poly(I:C) into $Mx1^{Cre}Myb^{flox/flox}$ (BM ablation) followed by transplantation of CD45.1 BM. **g** Representative flow-cytometry analysis showing expression of CD45.1 and CD45.2 on CD45+/lin−/CD11b+/F4/80+ macrophages in the adventitia and **h** percentage of CD45.1 macrophages after chimerism for 3 and 9 months is shown ($n = 3$ for 3 months; $n = 4$ for 9 months; seven individual experiments). **i, j** Pulse labelling of BM-derived cells in $Kit^{MCM}Rosa^{GFP}$ mice by tamoxifen feeding for 4 weeks. **i** Representative immunohistological images of aortas and **j** percentage of labelled blood monocytes ($n = 4$, 4 independent experiments) and adventitial macrophages ($n = 5$, 5 independent experiments). Scale bar as depicted in images. Two-sided $t$-tests was performed and mean ± SD is shown.

ablated. We then switched to normal chow diet, which resulted in recovery of arterial macrophages to near baseline values within 7 days, quantified by flow cytometry and histology (Fig. 4a–c).

Thus, this model allows to assess the kinetics of macrophage ablation and recovery following blockade of CSF1R signalling. To

compare the response of BM- and embryonic-derived macrophages to PLX5622, we applied the $Flt3^{Cre}Rosa26^{mTmG}$ reporter mice, in which constitutive expression of red (tomato) fluorescence switches to green (GFP) once the Flt3 promoter is active[27]. Both eYFP+ and eYFP− macrophage populations were depleted

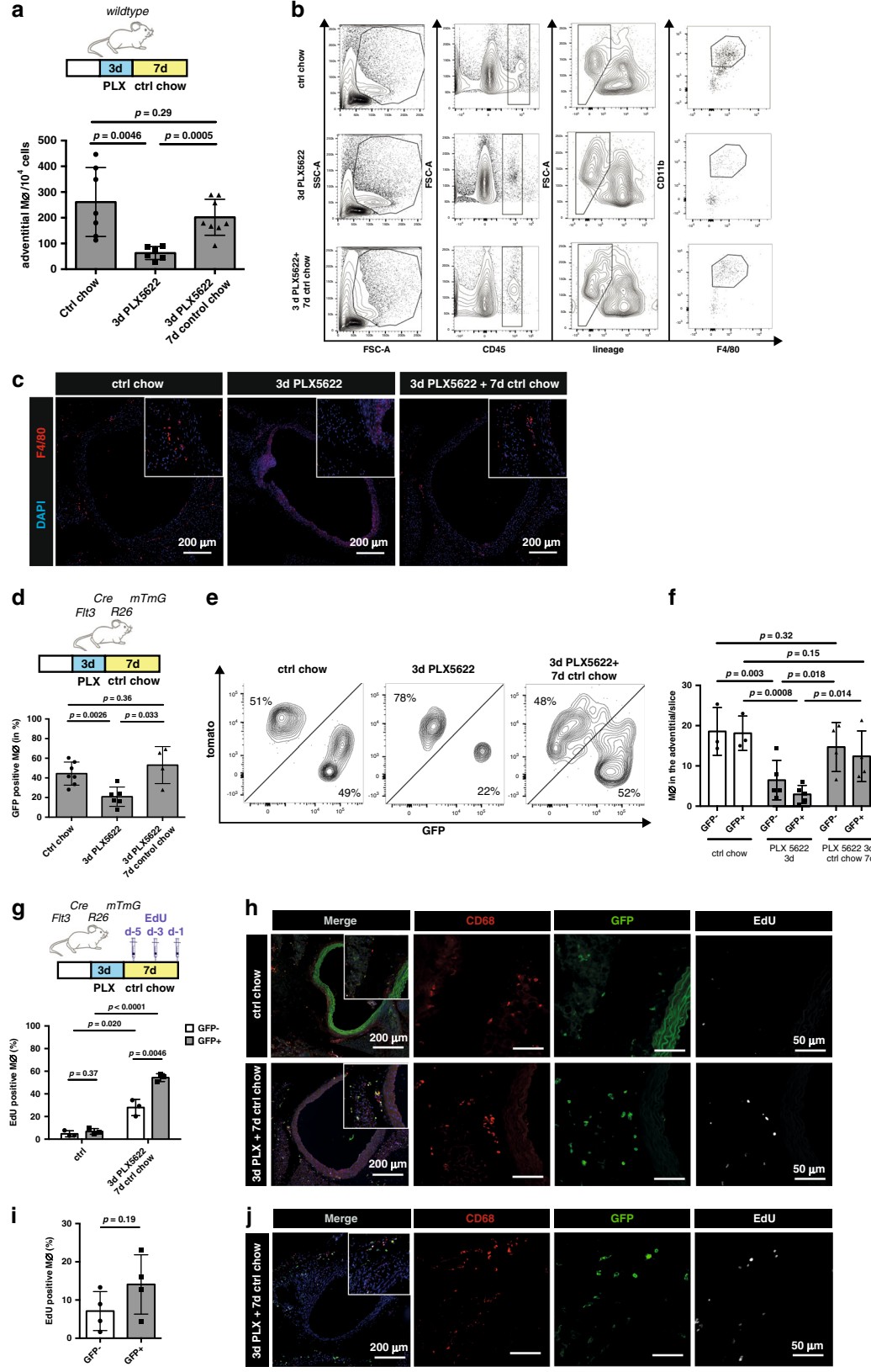

after 3 days of PLX5622 treatment (Fig. 4d–f). The relative proportion of eYFP+ macrophages was reduced from ~45 to ~20% indicating that the depletion of BM-derived macrophages was stronger (Fig. 4d, e).

When we switched the PLX5622-diet to normal chow after 3 days, not only macrophage numbers normalized within one week but also the relative proportion of BM-derived cells contributing to the pool of arterial macrophages was restored to baseline values of ~45% (Fig. 4d–f). Measurement of macrophage EdU incorporation during the recovery phase indicated strong proliferation of both GFP+ and GFP− populations (Fig. 4g, h). Whereas EdU incorporation in GFP− resident macrophages

**Fig. 4 BM-derived macrophages are more susceptible to Csf1r-inhibiton. a–c** CSF1R-inhibition in wildtype mice. **a** Schematic of macrophage depletion by CSF1R-inhibiton for 3 days (PLX5622 chow) followed by 7 days of control chow (recovery) and number of macrophages (Mφ) in the adventitia of the respective treatment group (n = 7 for ctrl chow, n = 6 for 3 days of PLX5622; n = 8 for 3 days of PLX5622 followed by 7 days of control chow; two individual experiments for each condition). **b** Representative FACS analysis of all adventitial macrophages isolated from individual mice fed with either control chow (upper panel), 3 days of PLX5622 (middle panel) or 3 days of PLX followed by 7 days of control chow (lower panel). Adventitial macrophages were gated on CD45+, lineage− (CD11C, Ly6G, TER119, Siglec-F, TCR-ß, Nk1.1), CD11b+, F480hi. **c** Representative immunohistology of macrophages in the adventitia (n = 3, three independent experiments). **d–j** CSF1R-inhibition in *Flt3CreRosa26mTmG* mice. **d** Percentage of GFP+ macrophages in the adventitia (n = 7 for ctrl chow, n = 6 for 3 days of PLX5622; n = 4 for 3 days of PLX5622 followed by 7 days of control chow; two individual experiments for each condition). **e** Flow cytometry of tomato and GFP expression in adventitial macrophages. **f** Histological quantification of GFP + CD68 + macrophages (n = 3 for ctrl chow, n = 5 for 3 days of PLX5622; n = 4 for 3 days of PLX5622 followed by 7 days of control chow; 2 individual experiments for each condition). **g, h** Macrophage proliferation after depletion by PLX5622 in *Flt3CreRosa26eYFP* mice and injection of EdU i.p. either **g** 5, 3, and 1 days before euthanization (n = 3 for ctrl chow, n = 3 for 3 days of PLX5622 followed by 7 days of control chow; 1 individual experiments for each condition) or **i** only once 2 h before euthanization (to determine local proliferation; n = 4; 1 individual experiment). **h, j** Representative immunohistological images from aortas. Scale bar as depicted in images. Two-sided *t*-tests was performed and mean ± SD is shown.

indicated their local proliferation, BM-dependent GFP+ cells could either proliferate locally or be recruited as proliferating cells from BM cavities. Additional experiments with short-term EdU treatment (single injection 2 h before euthanization) in *Flt³CreRosa26mTmG* mice confirmed the increased proliferation of BM-derived GFP+ macrophages and indicated that recruitment of circulating GFP+ cells may play a minor role for the restoration of the arterial macrophage pool under these conditions (Fig. 4i, j). In summary, proliferation of both BM as well as embryo-derived macrophages contributes to the recovery of depleted macrophages.

**Transient recruitment of BM macrophages in AngII inflammation.** As we have characterized the dependence of arterial macrophage populations on CSF1R homeostatic signalling, we next determined macrophage responses to inflammatory cues. Continuous application of AngII via an osmotic pump induces vascular inflammation increasing the number of tissue macrophages (Supplementary Fig. 9A, B)[14,28,29] and causing arterial fibrosis (Supplementary Fig. 9D, E)[30]. The AngII-induced increase in macrophages is largely confined to the adventitia, which harbors ~90% of macrophages (Fig. 5c, Supplementary Fig. 9C). In *Flt3CreRosa26eYFP* mice on AngII treatment, we observed a significant increase in BM-derived macrophages in the early phase of inflammation (Fig. 5a). To compare the proliferative capacity of the different macrophage populations, we evaluated EdU incorporation in arterial macrophages of *Flt3CreRosa26eYFP* mice on AngII treatment. Using repeated EdU injections, comparable to the regimen used at baseline conditions (Fig. 3e), proliferation was profoundly increased in acute inflammation compared to baseline in eYFP− arterial macrophages (Fig. 5b, c). As described above, incorporation of EdU in BM-derived macrophages may not solely represent local proliferation. Injection of EdU 2 h before euthanization mostly excludes the impact of monocyte-influx into the arterial wall and therefore depicts local proliferation[11]. Interestingly, using this labelling strategy, we found that proliferation of GFP+ macrophages was lower compared to GFP− resident macrophages (Fig. 5d, e).

To determine whether the increase of BM-derived macrophages was mediated by recruitment of BM-derived cells we implanted AngII pumps into BM chimeric *Mx1CreMyb*fox/flox; *CD45.2* which had been induced with poly(I:C) and transplanted with CD45.1 congenic BM cells. At day 10 of AngII treatment, we observed an increased CD45.1 chimerism of ~30% as compared to <10% in control mice receiving osmotic pumps with saline (Fig. 5f). Thus, the inflammatory response to AngII is characterized by local proliferation of embryo-derived arterial macrophages. The increase in eYFP+ macrophages in early

inflammation is largely driven by recruitment of BM-derived cells.

To test the macrophage response to chronic AngII inflammation, we applied AngII treatment over 4 weeks via osmotic pumps. Approximately 40% of arterial macrophages in *Flt3CreRosa26eYFP* mice were eYFP+ after 4 weeks, which was similar to the labelling in control mice receiving saline pumps or at steady-state conditions (Fig. 5a, Fig. 3h). In analogy to this experiment, the chimerism of transplanted CD45.1 cells in BM chimeric *Mx1CreMyb*flox/flox mice was not different between saline and AngII treatment after 4 weeks (Fig. 5f). Additionally, eYFP+ as well as eYFP− macrophages in *Flt3CreRosa26eYFP* mice showed similar low-level EdU incorporation 28 days after AngII treatment (Fig. 5b, c). These findings indicate that increased recruitment of BM precursors, which is a hallmark of acute inflammation, is only transient. They also suggest that the relative contribution of macrophages in the mouse aorta in relation to their developmental paths could be maintained in chronic inflammatory conditions.

**Profiling of adventitial immune cells in AngII inflammation.** Next, we explored the transcriptional landscape of macrophage populations in response to AngII-mediated inflammation at single-cell resolution (Fig. 6). In analogy to the steady-state analysis described above, we isolated total CD45+ immune cells from adventitias of *RankCreRosa26eYFP* mice after 10 days of AngII treatment. As demonstrated above, leukocytes are largely confined to the adventitia in AngII inflammation (Fig. 5c), and ~90% of macrophages are located in this layer of the aorta (Supplementary Fig. 9C). Unsupervised Seurat-based clustering of 4419 cells revealed 12 clusters of immune cells (Fig. 6a; Supplementary Fig. 10A–C, F, Supplementary Data 1). Macrophages continued to represent the most abundant immune cells in the arterial wall after 10 days of AngII application (Fig. 6c). Their gene expression patterns were more diverse than in steady state, allowing us to differentiate five macrophage populations (clusters 0, 2, 3, 4, and 7). In addition, we identified different clusters of dendritic cells (clusters 1,10, and 11), eosinophil (cluster 9), and neutrophils (cluster 5) in the inflamed aorta (Supplementary Fig. 11).

Macrophage cluster 2 contained most of EMP-derived (*Yfp+*) macrophages (Fig. 6b, c), and was characterized by expression of *Lyve-1*, *Stab1*, *Gas6*, *Cd163*, and *Pf4* (Fig. 6d, e) mirroring the homeostatic cluster identified in steady state. A relevant proportion of *Yfp−* BM-derived macrophages was also present in this cluster in response to AngII inflammation. The only other cluster containing a relevant proportion of *Yfp+* macrophages was cluster 0, which showed high expression of *CD72* as well as MHCII-related genes (*CD74, H2-Eb, H2-Ab1, H2-Aa,* and *Ciita*).

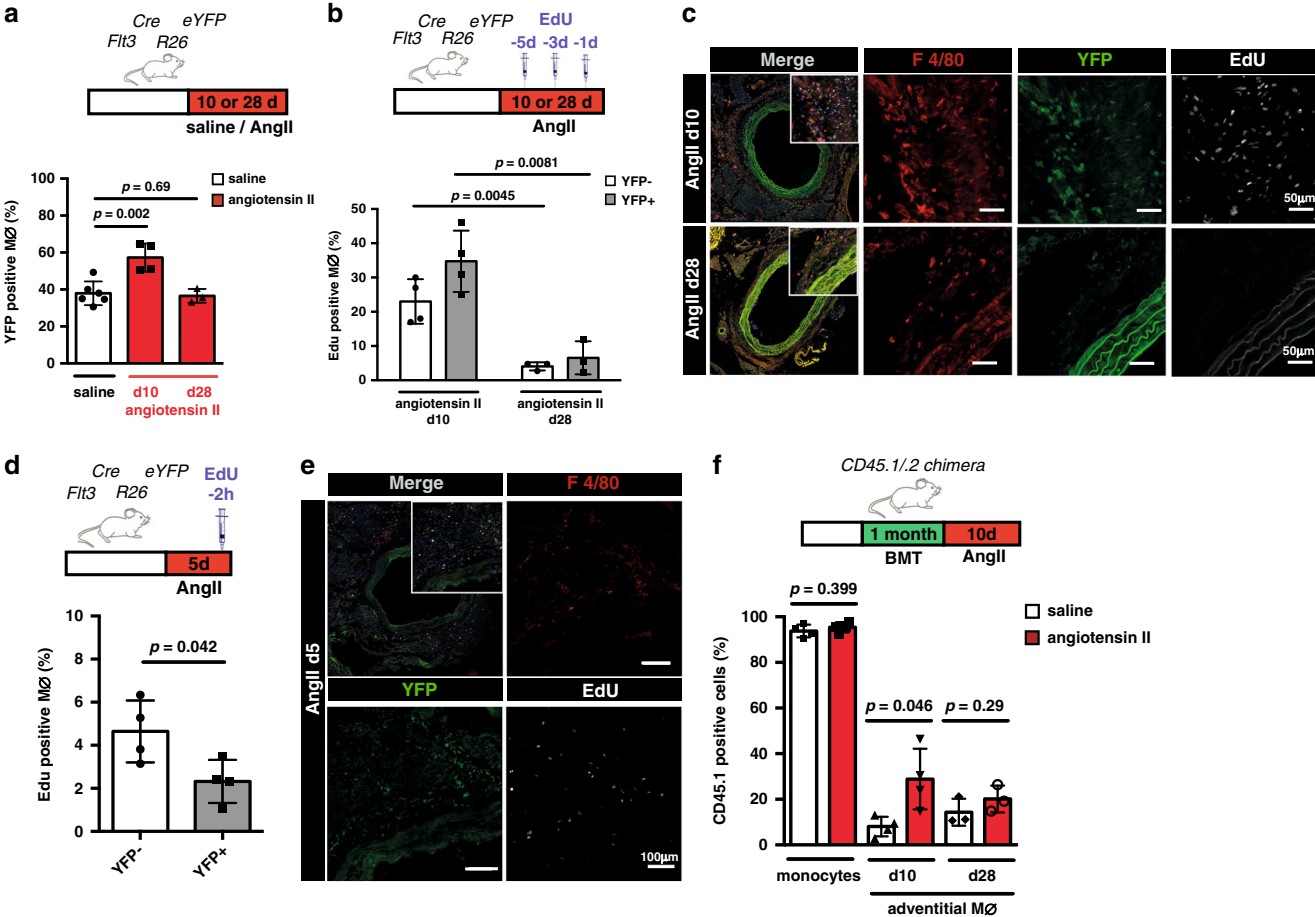

**Fig. 5 Local proliferation of resident macrophages and recruitment of BM-derived macrophages in response to AngII. a** Schematic of AngII delivery by osmotic pumps implanted into *Flt3^CreRosa26^eYFP* mice for either 10 or 28 days. Percentage of eYFP+ macrophages in the adventitia (*n* = 6 for steady state; *n* = 4 for 10d of AngII, *n* = 3 for 28d of AngII; three individual experiments for steady state and 10 days, and 1 individual experiment for 28 days of AngII). **b, d** schematic graphs analysing the proliferative behavior of macrophages in response to AngII by repeated injection of EdU i.p. 1, 3 and 5 days (**b**; *n* = 4 for 10d of AngII, *n* = 3 for 28d of AngII; each 2 independent experiment) or only once 2 h (**d**; *n* = 4 for 5d of AngII; 1 individual experiment) before euthanization. Quantification of proliferating eYFP− and eYFP+ macrophages in respective groups is shown. **c, e** representative immunohistology of proliferating arterial macrophages in response to AngII. **f** AngII or saline pumps were implanted into poly(I:C) treated BM chimeric CD45.2 *Mx1^CreMyb^flox/flox* recipient mice transplanted with CD45.1 BM. Bar-graph shows percentage of CD45.1 monocytes in the blood after 10 days of AngII treatment and CD45.1 macrophages in the adventitia after 10 days (*n* = 4 for each group; 2 individual experiments) and after 28 days (*n* = 3 for each group; each two independent experiments). Scale bar as depicted in images. Two-sided *t*-tests was performed. Mean ± SD is shown.

This cluster did not express cluster-specific signature genes compared to the other macrophage clusters, as highlighted in the heatmap analysis, and they were therefore termed Mixed Macrophages (Fig. 6d, e). Clusters 3, 4, and 7 represented BM-derived macrophages and displayed profound heterogeneity. While cluster 3 was defined by high expression of *Spp1, Arg1 Chil3,* and *Tgfß*, genes linked to reparative functions, cluster 7 showed high levels of proinflammatory genes, e.g., interferone-induced genes (*Ifit1, Ifi211, Ifit2, Ifit3,* and *Irf7*) and was the only cluster expressing high levels of *Ly6c*. Cluster 4 was characterized by high expression of *Ear2* and *Retnla*, which are associated with type-2 cytokine responses, and the proliferation marker *Mki67* (Fig. 6d, e). We further evaluated biological pathways enriched in respective macrophage populations by analysing gene ontology terms associated with differentially expressed genes using ClueGO (Supplementary Fig. 10E). These data confirm the broad functional spectrum of arterial macrophages and their heterogeneity seen in other models of vascular disease[3,4,31].

It should be noted that a potential limitation of our scRNA-seq analysis in AngII conditions is the relatively low coverage. Focusing the scRNA-Seq analysis on the adventitial tissue of the

aorta, thereby excluding intima and media, can be considered another limitation. However, we analyzed most leukocytes that accumulated in the aorta in AngII inflammation as demonstrated by extensive histology.

**Distinct transcriptional patterns in EMP-derived macrophages.** In addition to the differential analysis of macrophage clusters provided in Fig. 6, we also compared the transcriptional profiles of all *Yfp+* (EMP-derived) and *Yfp−* macrophages in response to AngII inflammation (Fig. 7a, Supplementary Data 1). *Yfp+* macrophages expressed high levels of homeostasis-related genes, which is in-line with their main representation in cluster 2 (Fig. 6b). These homeostasis-related genes associated with biological pathways involved in VEGF production, negative regulation of apoptotic processes and regeneration (Fig. 7a). Additionally, we analysed pathways associated with secretory function of macrophages. In inflammatory conditions, EMP-derived macrophages expressed high levels of chemokines attracting other immune cells, such as monocytes, neutrophils, and lymphocytes (*Ccl2, Ccl7,* and *Ccl12*). Further, genes of the complement cascade (*C1qa, C1qb,* and *C1qc*) (Fig. 7b) were

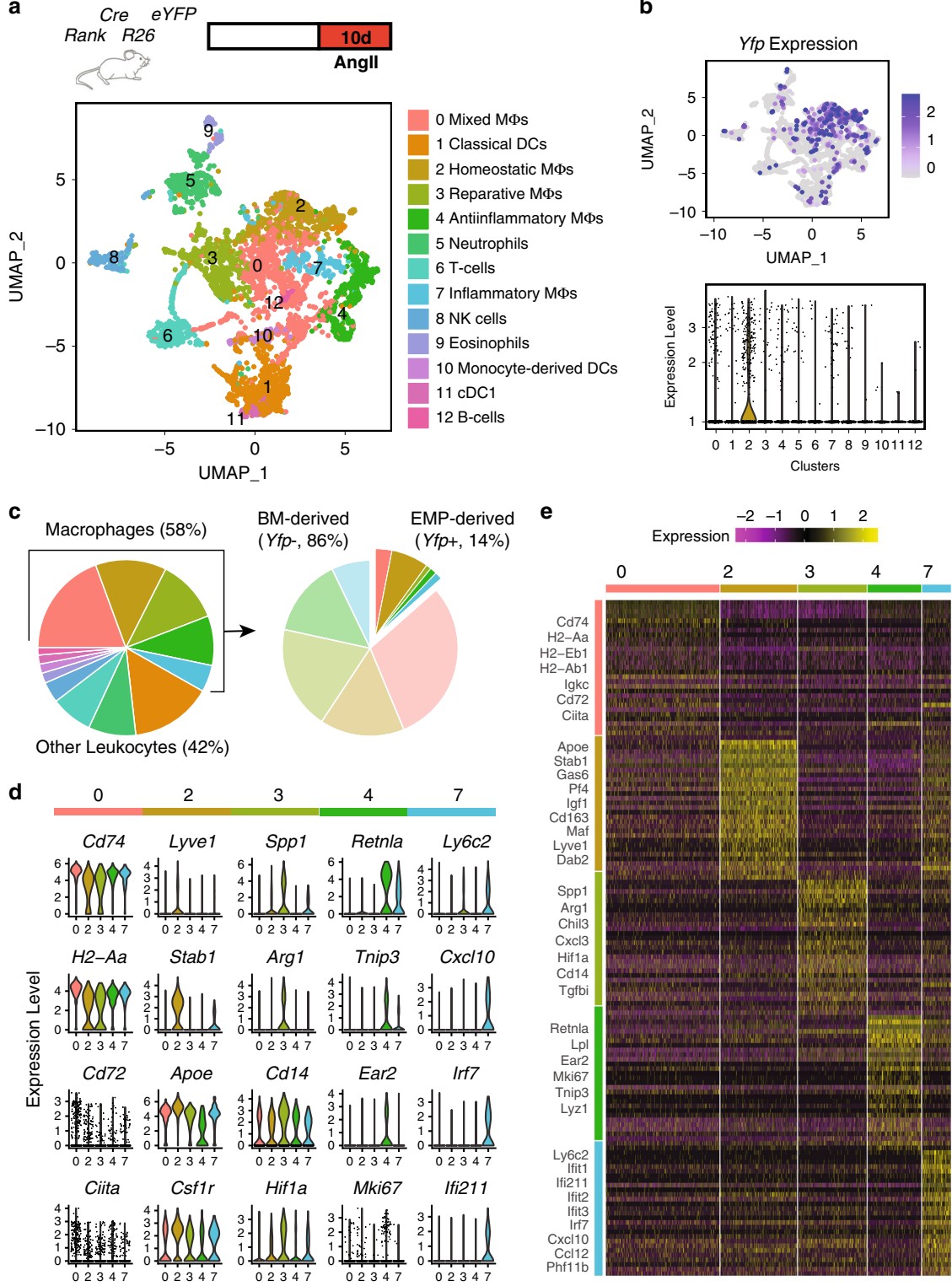

**Fig. 6 Transcriptional profiling of macrophages in response to AngII.** Single-cell transcriptome analysis reveals the heterogeneity of leukocyte populations in inflammation. **a** UMAP visualization of gene expression in single CD45+ cells in adventitias of *Rank*^CreRosa26^eYFP mice treated with AngII for 10 days (*n* = 4419 cell from 2 animals), depicting 13 different cell types including 5 distinct macrophage (Mϕ) subtypes. **b** Feature (upper panel) and Violin (lower panel) plots showing the expression of *Yfp* in adventitial leukocytes. Expression of *Yfp* is mainly identified in the cluster of homeostatic macrophages (cluster 2). **c** Proportion of different leukocyte populations in inflamed aorta (left panel) and fraction of *Yfp*-expressing cells within adventitial macrophages (right panel). **d** Violin plots representing significant (*P* value < 0.05) expression of markers of mixed (cluster 0), homeostatic (cluster 2), reparative (cluster 3), anti-inflammatory (cluster 4), and inflammatory (cluster 7) macrophages. **e** Heatmap showing top30 marker genes in each macrophage subset. Expression scale is Log2 fold-change of gene expression in the corresponding cluster against all other macrophage clusters. Representative markers of each subtype are labeled.

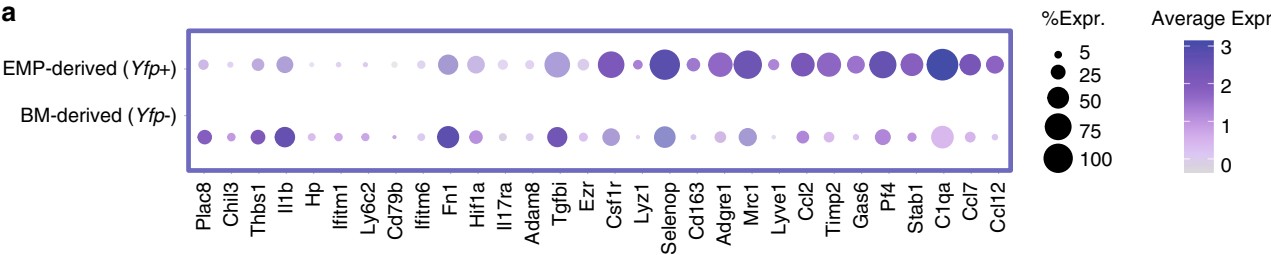

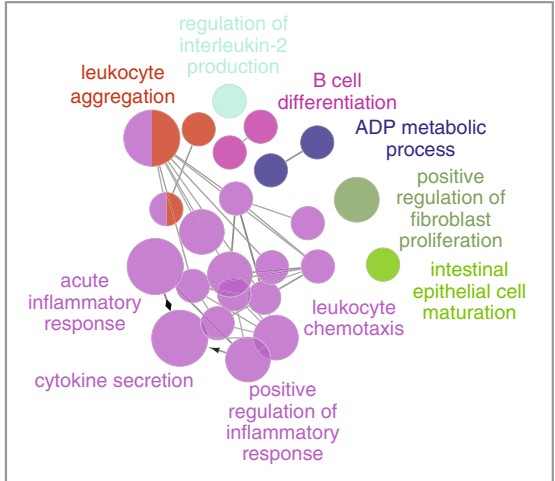

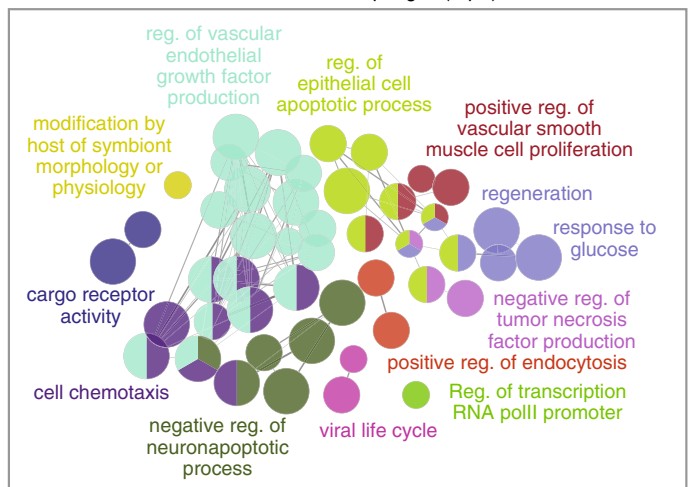

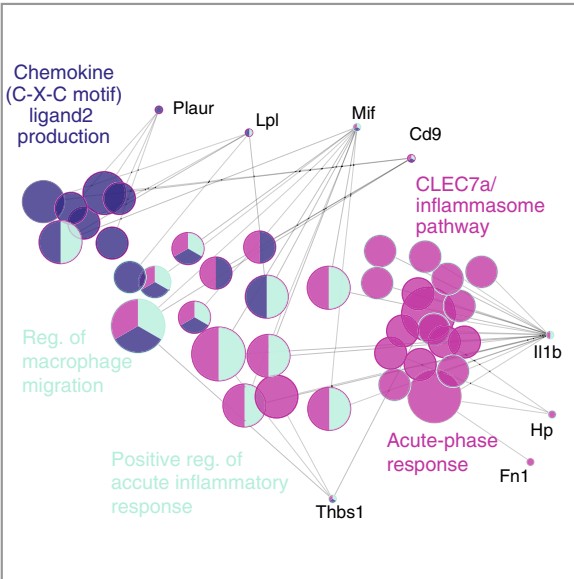

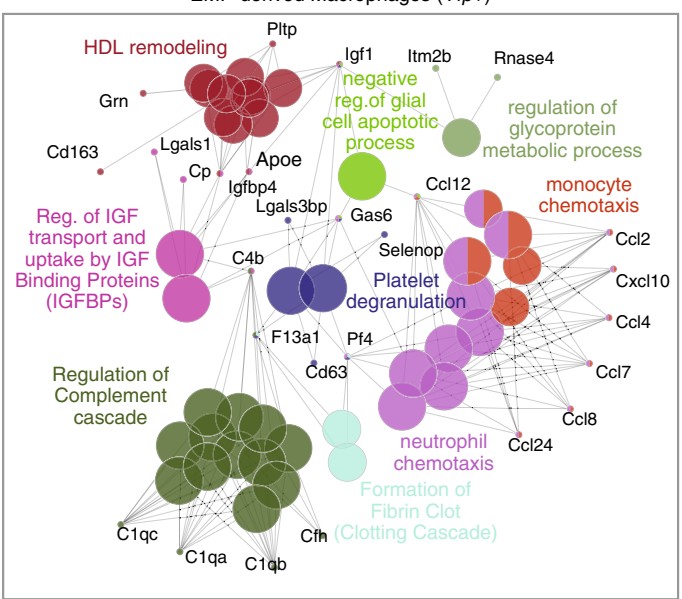

**Fig. 7 Biological network analysis of genes associated with macrophage clusters.** GO term and pathway analysis of EMP- and BM-derived macrophages in response to AngII. **a** Dot plot showing the top 15 genes, which are differentially expressed between EMP- and BM-derived macrophages (upper panel). Functionally grouped networks (lower panels) depicting biological processes and pathways associated with genes that are significantly upregulated in *Yfp+* or *Yfp−* macrophages (*p* value < 0.05, Log2FC > 0.25). Each node represents a specific biological process or pathway, and edges indicate the relations between different terms based on their kappa score level (>0.4). Nodes of the same color belong to functionally similar terms or pathways. The node size represents the term enrichment significance (*P* value < 0.0005 to <0.05). **b** Networks of functionally grouped pathways and processes associated with secretory markers of *Yfp+* or *Yfp−* macrophages. Genes with capability of producing secretory proteins were shortlisted from all significantly upregulated genes in each population and subjected to pathway enrichment analysis. The marker genes are identified using Seurat FindMarker function based on Wilcoxon Rank Sum test. ClueGo performs the enrichment test based on the hypergeometric distribution and Bonferroni correction.

upregulated in *Yfp+* macrophages. On the other hand, *Yfp−* macrophages expressed genes associated with cytokine secretion linking them to proinflammatory (e.g., *Il1b* and *Thbs1*) as well as anti-inflammatory processes (e.g., *Chil3* and *Plac8*) (Fig. 7a, b, Supplementary Fig. 10E).

Next we determined differential gene expression between EMP-derived and BM-derived macrophage lineages within the same cluster, as well as overlapping gene expression between clusters. Interestingly, *Yfp+* and *Yfp−* macrophages of the same cluster exhibited some differences in their gene expression (Supplementary Fig. 12). While differences were rather low in cluster 2 (homeostatic macrophages), they were more prominent in others such as cluster 7 (inflammatory macrophages). The presence of distinct gene signatures within clusters supports the notion that ontogeny impacts on macrophage gene expression and potentially function.

There was small overlap of genes between clusters (Supplementary Fig. 13). In the lineage of EMP-derived macrophages, we identified *Cx3cr1* to overlap between clusters 0, 2, and 7. Interestingly, chemokine (C-C motif) ligand 12 (*Ccl12*, MCP-5) was shared between all clusters of *Yfp+* macrophages in AngII inflammation. MCP-5 has previously been identified in activated macrophages[32]; however, its functional role is incompletely understood and could represent in interesting molecule to address in future studies.

In summary, the ontogeny of arterial macrophages is associated with defined biological pathways. EMP-derived macrophages are abundant in the adventitia from postnatal life into adulthood providing key homeostatic functions both in steady state and in response to AngII inflammation.

## Discussion

Macrophages are abundant in all tissues, including arteries, where they contribute to the maintenance of tissue homeostasis and act as effectors of innate immunity. Their activation and proinflammatory polarization are associated with arterial diseases[33,34]. Targeting of macrophages could potentially reduce tissue inflammation and impact positively on remodeling processes. Therefore, a thorough understanding of macrophage development, homeostasis and response to inflammatory signals in their tissue of residence is warranted.

We carried out an in-depth analysis of arterial macrophages at single-cell resolution in steady state and AngII inflammation in relation to their developmental paths. First, we revisited macrophage ontogeny harnessing a variety of complementary lineage-tracing models to map their origin in the mouse aorta. The initial hematopoietic site giving rise to macrophage progenitors is the YS[35]. YS-derived EMPs are a common source for tissue macrophages that persist for long periods of time (at least 45 weeks) in many organs such as brain, liver, and skin[6]. In other tissues they are replaced continuously or progressively by BM-derived hematopoietic cells[12,36]. In arteries, it has been suggested that tissue macrophages mainly derive from YS progenitors before birth, and that the majority of macrophages are replaced by BM-derived macrophages in early postnatal life[10]. Our findings are in-line with the dual origin of arterial macrophages from both YS and BM hematopoiesis. By analyzing adventitial macrophages in the aorta of *Tie2*[MerCreMer]*Rosa26*[eYFP] fate-mapping mice as well as by quantifying their total number numbers in *Rank*[Cre]*Rosa26*[eYFP] mice in a time course analysis, we determined that EMP-derived macrophages represent a major population in adulthood. Their number increases in postnatal life independently of BM hematopoiesis, reaches a maximum in adult mice but are then diminished in the process of ageing. The fraction of BM-derived macrophages increases in adolescence and remains stable in the further course of life. The relative increase of BM-

derived macrophages evident in aged mice is, therefore, driven by the exhaustion of resident EMP-derived macrophages during ageing rather than their replacement by BM-derived cells. The mechanisms underlying macrophage maintenance and survival in ageing remain to be explored. Potentially, alterations of the environment as well as changes on the cellular level of immune cells, including mitochondrial dysfunction and metabolic stress, could contribute to this process[37].

Local proliferation in tissues is an important macrophage feature, allowing them to maintain their populations but also to adapt to changing conditions[38]. In the first week of life, Langerhans cells in the epidermis undergo a boost of local proliferation[39]. Later on, these macrophages are maintained independently of BM HSCs[40]. Accordingly, EMP-derived adventitial macrophages substantially increase in numbers in postnatal life, whereas the number of BM-derived macrophages rises slowly over time. Together, EMP-derived macrophages are maintained by local proliferation and represent the predominant macrophage population in adult life.

We observed limited de novo contribution of blood monocytes to the pool of adventitial macrophages in healthy mice, as demonstrated by macrophage analysis in a *Myb*-dependent model of BM transplantation as well as using *Rank*[Cre]*Rosa26*[eYFP] and *Kit*[MerCreMer]*Rosa26*[GFP] fate-mapping models. This is in contrast to other BM-chimera models that rely on HSC ablation by irradiation. However, radiation induces tissue inflammation and can impair macrophage self-renewal capacity, thus promoting the replacement of arterial macrophages by HSC-derived cells[10]. Even in sustained inflammation in atherosclerotic arteries, local proliferation rather than monocyte recruitment increases lesional macrophage number[41].

We compared macrophage lineages in *Flt3*[Cre]*Rosa26*[eYFP] fate-mapping mice, in which macrophages displayed differences in their proliferative response to CSF1 and AngII. The chemokine CSF1 provides essential homeostatic signaling for the growth and differentiation of macrophages[8,25]. To evaluate the contribution of the BM to the recovery of the arterial macrophage pool after cessation of CSF1R-inhibition, a BM-chimera model was implemented. In-line with other studies using irradiation dependent chimera models[10], they found substantial replacement of resident by BM-derived macrophages already in steady state and near-complete replacement after recovery from CSF1R-mediated macrophage depletion. To circumvent the bias of irradiation and its described consequences, we used the Flt3-reporter mice to determine the contribution of BM-derived macrophages to macrophage recovery. When we interfered with CSF1R signalling, BM-derived arterial macrophages displayed a stronger ablative response compared to their EMP-derived counterparts. A similar effect of CSF1R-inhibition has been reported in the lung, where BM-derived interstitial macrophages show increased depletion compared to embryo-derived alveolar macrophages[42]. After acute ablation, arterial macrophages recovered quickly and numbers returned to baseline within 7 days. We found that the recovery of the tissue macrophage pool was driven by local proliferation of embryonic and also BM-derived cells, when proliferation capacity of arterial macrophages was not affected by radiation. Taken together, we provide evidence that survival and proliferation of both BM- and EMP-derived macrophages is driven by CSF1R signalling, identifying it as an essential factor of macrophage homeostasis in the arterial wall.

The macrophage response to AngII inflammation was also characterized by induction of local proliferation. The increase in macrophage numbers in the early phase of AngII application (<10 days) is driven by recruitment of blood monocytes, as shown in *Flt3*[Cre]*Rosa26*[eYFP] mice and the *Myb*-dependent BM-chimera model. This process can be targeted therapeutically by blockade of CCR2 signaling, which abrogates inflammatory monocytes in the blood circulation and decreases macrophage numbers in the

arterial wall, leading to beneficial effects on fibrotic remodeling and blood pressure[14]. In the subacute phase of AngII exposure (30 days), neither substantial recruitment of blood monocytes nor proliferation of both macrophage populations was sustained, suggesting that the early phase of inflammation (<10 days) is critical in this model to drive arterial injury.

The differences in the ontogeny of macrophage populations and their responses to AngII inflammation led us to address the heterogeneity of arterial macrophages in more detail. We therefore carried out scRNA-seq of adventitial immune cells[3,4]. We show here that macrophage ontogeny is associated with the expression of distinct sets of genes in the transcription analysis, which are associated with specific biological functions. Given the important role of the environment in programming macrophage phenotype and functions[43,44], the presence of different macrophage clusters is intriguing. Even more so, since eYFP+ and eYFP− macrophage populations showed no specific orientation throughout the aorta in $Flt3^{Cre}Rosa26^{eYFP}$ mice, substantiating the notion that they inhabit the same niche.

EMP-derived macrophages are characterized by high expression of Lyve-1, Stab1, Gas6, and Cd163, among other genes. The hyaluronan receptor LYVE-1 has been identified as a key marker of resident macrophages in cardiovascular tissues[19,45]. In arteries, LYVE-1 exerts important homeostatic functions by controlling collagen expression in vascular smooth muscle cells[19]. Further, LYVE-1 provides pro-angiogenic functions[18]. Absence of LYVE-1 expressing macrophages results in arterial remodeling and stiffness, and consequently in blood pressure dysregulation[19]. Growth arrest-specific protein 6 (GAS6)[22] as well as Stabilin-1 (STAB1)[20] are efferocytosis ligands, which serve as scavenger repectors and participate in the clearance of apoptotic cells. Macrophage-specific deficiency in Stab1 exacerbates adverse remodeling in liver fibrosis due to reduced clearance of metabolic products[20]. CD163 also represents a scavenger receptor commonly expressed on arterial macrophages and is associated with anti-atherosclerotic functions[46,47]. Taken together, these findings provide a strong link between EMP-derived arterial macrophages and homeostatic functions in arteries.

Chil3/Arg1 and Ear2/Retnla, representing clusters 3 and 4, respectively, are of particular interest since these genes are associated with anti-inflammatory and reparative functions of macrophages[48]. They are induced by type-2 cytokines such as IL-4[49], and are specifically upregulated by apoptotic cells at sites of tissue injury[50]. Chil3/Arg1 expressing macrophages have been associated with repair mechanisms in organ injury[51], including cardiovascular tissues[52]. Erba-related protein 2 (Ear2) encodes for the nuclear receptor Nr2f6, which functions as an immune checkpoint in T cells[53]. A macrophage cluster expressing high levels of both Ear2 and Retnla was recently identified in atherosclerotic arteries[31]. In-line with the latter report, we identified Ear2/Retnla to represent a specific cluster of macrophages (cluster 4). We show here that both clusters 3 and 4 are mainly derived from BM hematopoietic precursors. In the lung, Retnla expressing macrophages protect from pathogen injury and promote the resolution of inflammation[54]. Their functions in cardiovascular tissues remain to be explored.

In summary, arterial macrophages are of heterogenous nature. Their different developmental origins contribute to this heterogeneity, and are associated with distinct patterns of gene expression and differential responses to homeostatic signals and inflammatory cues. EMP-derived BM-independent macrophages represent a major population of macrophages in healthy adult mice providing anti-inflammatory and homeostatic functions. They are diminished in ageing due to yet undefined mechanisms, and their loss is not compensated by BM hematopoiesis. In AngII inflammation, BM-derived macrophages invade the inflamed adventitial tissue and

show heterogenous transcriptional profiles linking them to inflammatory as well as reparative functions. Resident EMP-derived macrophages respond by upregulation of gene pathways linked to tissue regeneration, chemotaxis, apoptosis regulation and modulation of distinct secretory proteins. Our findings contribute to the understanding of macrophage heterogeneity and functions in relation to their ontogeny. Differential gene expression and functions of macrophage populations may be conducive to development of targeted strategies in inflammatory conditions.

## Methods

**Mice.** $Flt3^{Cre55}$, $Myb^{-/-56}$, $Myb^{fl/fl57}$, $Mx1^{Cre58}$ (Stock No: 003556), $Csf1r^{MerCreMer59}$, $Tie2^{MCM6}$, $c$-$Kit^{MerCreMer60}$, $Rank^{Cre}$ (also known as $Tnfrsf11a^{Cre}$)[17,61], $Rosa26^{mTmG27}$ (Stock No: 007676), $Rosa26^{eYFP62}$ (Stock No: 006148), and $Rosa26^{eGFP60}$ reporter mice have been previously described. PCR genotyping was performed according to protocols described previously. All primers used are described in Supplementary Table 1.

Congenic C57BL/6 CD45.1 (Ly5.1; Stock No: 006584) mice were used as bone marrow donors. $Csf1r^{MCM}$ were on FVB background, other mice were on C57BL/6 (CD45.2) background. All experiments included littermate controls and the minimum sample size used was three. For fate-mapping analysis of Flt3+ precursors, $Flt3^{Cre}$ males (the transgene is located on the Y chromosome) were crossed to homozygous $Rosa26^{eYFP}$ or $Rosa26^{mtmg}$ reporter females. $Flt3^{Cre}$ males were used for lineage-tracing experiments and female littermates served as Cre-negative controls. For all other experiments offspring of both sexes were used. Embryonic development was estimated considering the day of vaginal plug formation as 0.5 days post-coitum (dpc), and staged by developmental criteria.

$Kit^{MerCreMer}$ were housed in a barrier facility, all other mice were maintained in a specific pathogen-free environment. All experimental animals were co-housed and fed standard chow diet ad libitum. Animals were 12–16-week-old mice except for time course experiments, in which mouse age is indicated (Figs. 1a, b, e–g, 3a–d).

We have complied with all relevant ethical regulations. Animal studies were approved by the local regulatory agency (Regierung von Oberbayern, Munich, Germany), record numbers 55.2.1.54-2532-93-13, 55.2-1-54-2532-55-15, 55.2.1.54-2532-183-16, ROB-55.2-2532.Vet_02-19-17, and ROB-55.2-2532.Vet_02-19-1.

**Processing of adult tissues for flow cytometry.** Adult mice were euthanized by cervical dislocation under anaesthesia. Blood was collected by cardiac puncture from anaesthetized mice (fentanyl (0.05 mg/kg per kg), midazolam (5.0 mg/kg per kg), and medetomidine (0.5 mg/kg per kg)). Under anaesthesia, mice were perfused by gentle intracardiac injection of 10 ml heparinised PBS. The mouse aorta was harvested using a dissecting microscope. Enzymatic digestion with Collagenase II (Worthington Biochemical; 20 mg/ml) and Elastase (Worthington Biochemical; 20 mg/ml) was carried out at 37 °C for 15 min, then the adventitia was separated from the media and passed through a 70 μm filter. After 15 min incubation with purified anti-CD16/32 (FcγRIII/II; 1/50), antibodies were added and incubated for 30 min. All flow-cytometry studies were performed using a BD Biosciences FACSCanto II or BD FACS LSR Fortessa. After exclusion of doublets, macrophages in the aorta were characterized as CD45+ lin− (CD11C, Ly6G, Ter119, Siglec-F, TCR-ß, and Nk1.1) CD11b+ F4/80+ cells.

Bone marrow cells were flushed from femurs, tibias and coxa using PBS supplemented with 5% heat-inactivated FCS in PBS. Cells were filtered through a 20-μm filter (Falcon). Livers were digested with collagenase IV (600U/ml) and DNase I (2,5 g/ml) at 37 °C in RPMI for 45 min, washed and resuspended in HBSS. NPC were isolated by Percoll gradient (50%/25%) centrifugation at $1800 \times g$, 15 min. The NPC-containing middle layer was collected and washed. Fc receptors were blocked by incubating cells in 5% FCS with purified mouse IgG (500 mg/ml, Jackson ImmunoResearch Laboratories). All stainings were performed in 5% FCS on ice for 30 min with optimal dilutions of commercially prepared antibodies. Dead cells were excluded by staining with Sytox Blue (Invitrogen). Cells were analysed on a FACSFortessa.

All data were analysed using FlowJo 10.07 (Tree Star). All antibodies used are listed in the Supplementary Table 2.

**FACS sorting and cytospin for embryonic tissues.** Mice were euthanized by cervical dislocation under anaesthesia. Embryos at embryonic day E16.5 were dissected out from the uterus and washed in 4 °C phosphate buffered saline (PBS, Invitrogen). To obtain single-cell suspensions, the aorta was incubated in PBS containing 1 mg/ml Collagenase D (Roche), 100U/ml Desoxyribonuclease (DNAse I, Sigma) and 3% fetal bovine serum (Invitrogen) at 37 °C. Finally, the tissue was mechanically dissociated and passed through a 100μm cell strainer to obtain single-cell suspensions.

Arterial cells from YFP+F4/80bright cells from E16.5 pulsed at E8.5 were FACS-sorted into FCS-coated tubes for cytospin preparations. Sorted YFP+ F4/80bright cells were May Grünwald–Giemsa stained and analysed using light microscopy (Carl Zeiss).

Cell sorting to obtain the whole immune cell population of adventitias of $Rank^{Cre}Rosa26^{eYFP}$ mice for single-cell analysis was performed on a MoFlo Astrios (Beckman Coulter). CD45-positive cells were enriched from adventitias by using magnetic beads and LS columns (CD45 MicroBeads; Miltenyi Biotec). Cells were then sorted as live CD45 cells from mice. Dead cells were identified with SYTOX Orange Dead Cell Stain.

**Single-cell RNA sequencing and analysis**. After sorting of adventitial CD45-positive immune cells from whole aortas (aorta ascendens to aortic bifurcation), viable cells were proceeded for single-cell capture, barcoding, and library preparation using Chromium Next GEM single-cell 3' (v3.1, 10x Genomics) according to manufacturer's specifications. Following, pooled libraries were sequenced on a Illumina HiSeq1500 sequencer (Illumina, San Diego, USA) in paired-end mode with assymetric read length of 28 + 91 bp and a single indexing read of 8 bp.

The reads were demultiplexed using Je-demultiplex-illu[63] and mapped against a customized mouse reference genome (GRCm38.p6, Gencode annotation M24) including eYFP sequence using CellRanger (v3.1.0, 10x Genomics). The filtered feature-barcode matrices passed quality-control steps of CellRanger and were subsequently analysed using the R package Seurat (v3.1)[64]. To ensure retaining of only high-quality cells, cells with <300 genes or >6500 detected genes or >5% of mitochondrial reads were omitted from the downstream analysis. SCTransform[64] function was used for normalization and scaling of raw counts and regressing out the unwanted sources of variation (such as mitochondrial gene content and cell-cycle stage). SCTransform also yields the highly variable features that were used for linear dimensional reduction using principle component analysis (PCA). The components that contribute significantly to the dimensionality of the data were identified using JackStraw test and used for unsupervised graph-based clustering (resolution 0.4) and Uniform Manifold Approximation and Projection (UMAP) embedding and visualization.

The significant marker genes of each cluster were identified using Seurat FindAllMarkers function and employed to assign cell types to each cluster. The top 100 cluster markers were used to determine the biological processes associated with each cluster using ClueGO application (v2.5.6)[65] ran within Cytoscape (v3.8.0) [66]network analysis tool.

Differentially expressed genes (>50% expression change, $P$ value < 0.05) between $Yfp+$ and $Yfp-$ subpopulations were detected with FindMarkers function of Seurat package based on implemented Wilcox test and UniProt/Swiss-prot[67] database was used to determine secretory protein markers of each population. Enriched pathways associated with these secretory protein markers were identified using CluePedia (v1.5.6)[68] and the integrated pathway databases KEGG, REACTOME and WikiPathways.

Genes that were differentially expressed between $Yfp+$ and $Yfp-$ macrophages within different clusters, were identified using Seurat FindMarker function. Venn diagrams were drawn from significant genes ($P$ value < 0.05) using the web tool at http://bioinformatics.psb.ugent.be/webtools/Venn/. Volcano plots for visualization of differentially expressed genes were generated using R package EnhancedVolcano.

**Pulse labelling**. For genetic cell labelling we crossed tamoxifen-inducible $Csf1r^{MCM}$, $Tie2^{MCM}$, and $Kit^{MCM}$ with $Rosa26^{eYFP}$, $Rosa26^{GFP}$, or $Rosa^{mTmG}$ reporter mice. In $Csf1r^{MCM}$ embryos recombination was induced by single injection at E8.5 of 75 µg per g (body weight) of 4-hydroxytamoxifen (Sigma) into pregnant females. The 4-hydroxytamoxifen was supplemented with 37.5 µg per g (body weight) progesterone (Sigma) to counteract the mixed oestrogen agonist effects of tamoxifen, which can result in fetal abortions. In $Tie2^{MCM}Rosa26^{eYFP}$ embryos recombination was induced by treatment of pregnant females by gavage at different timepoints at E7.5 and E10.5 with a single dose of 2.5 mg tamoxifen (Sigma) and 1.25 mg progesterone. In $Kit^{MCM}$ animals recombination was induced in 6–8-week-old adult animals using oral tamoxifen feeding (400 mg/kg diet, which approximates to a daily dose of 40 mg/kg mouse body weight; purchased from Envigo).

**Analysis of cell proliferation**. To assess proliferation under different conditions, 5–ethynyl–2–deoxyuridine (EdU) was administered i.p. at different timepoints as described in the manuscript (1 mg in 150 µl).

**BM transplantation after conditional deletion of Myb**. The $Mx1^{Cre}$ mouse strain carrying an interferon inducible Cre-cassette[58], was used to conditionally delete the floxed Myb gene in adult $Myb^{flox/flox}$ mice. Depletion was performed as previously described[9,24]. To substitute for the depletion of hematopoietic cells, mice were transplanted with $1 \times 10^6$ CD45.1 $Myb^{+/+}$ BM cells at 24 h after the last of four poly(I:C) injections. When no sufficient chimerism was reached after initial transplantation retreatment with poly (I:C) was performed. Only mice with a chimerism of >90% were included into experiments.

**Osmotic pumps**. Alzet pumps were used for subcutaneous application of 1.8 µg AngII/kg/min or saline for different time periods (5, 10, and 28 days). The back of mice was shaved and disinfected using iodine solution. Subcutaneous implantation via a small incision in the back was performed under general anaesthesia using

fentanyl (0.05 mg/kg per kg), midazolam (5.0 mg per kg), and medetomidine (0.5 mg/kg per kg).

**Macrophage depletion**. To deplete arterial macrophages, we used the selective Csf1r-inhibitor PLX5622. Control and PLX5622 (300 ppm formulated in AIN-76A standard chow, Research Diets, Inc.) chows were kindly provided by Plexxikon Inc (Berkeley, CA).

**Analysis of macrophages in aorta sections**. After cervical dislocation under terminal anaesthesia, mice were perfused by intracardiac injection of 20 ml 4% PFA. Arterial tissues were carefully removed and washed with PBS and transferred overnight into 30% sucrose in PBS. Samples were then embedded in Tissue-Tek OCT (Sakura Finetek) and cryoblocks were sliced into 12-µm sections.

Cell numbers were assessed by systematic counting of the aorta using a fluorescence microscope equipped with an ×60 oil immersion objective (Carl Zeiss). Counting was performed in a blinded fashion. For imaging, a confocal laser scanning microscope (LSM 510, Carl Zeiss) was used and images were processed using Adobe Photoshop for adjustment of contrast and size. Detection of EdU was performed in accordance to the manufacturers protocol on cryosections of the aorta. All antibodies used are listed in the Supplementary Table 2.

**Statistical analysis**. We used the Student's $t$-test (Prism GraphPad). Welch's correction for unequal variances was used when applicable. A $p$ value of $p < 0.05$ was considered significant. All data are presented as mean ± standard deviation. All measurements were taken from distinct samples.

**Reporting summary**. Further information on research design is available in the Nature Research Reporting Summary linked to this article.

## Data availability

The authors declare that all data supporting the findings of this study are available within the paper and its supplementary information files. The scRNA-seq raw data have been deposited in NCBI GEO under the accession codes GSE155569, GSM4706334, and GSM4706335. There are no restrictions regarding data availability. Source data are provided with this paper.

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

## Acknowledgements

We thank Nicole Blount and Beate Jantz for animal husbandry. We thank Elisabeth Raatz, Cuong Kieu and Michael Lorenz for mouse genotyping. We thank H.-R. Rodewald for providing the Tie2$^{MCM}$Rosa26$^{eYFP}$ mice. This study was supported by the SFB 1123, projects A07 (C.Schu.) and B06 (S.M.), and the SFB914, projects A10 (C.Schu.), B02 (S.M.), and Z01 (H.I.-A.), as well as the DZHK (German Centre for Cardiovascular Research) and the BMBF (German Ministry of Education and Research) (grants 81Z0600204 to C.Schu., 81X2600252 to T.W. and 81X2600256 to M.F.). C.Str. was supported by a Gerok position of the SFB914. M.F. is supported by the DFG-funded Clinician Scientist Program *PRIME*. S.M. was supported by the Leducq Foundation Transatlantic Network "Clonal hematopoiesis and atherosclerosis". C.W. was supported

by FOR2033-A03, TRR127-A5, WA2837/6-1, and WA2837/7-1. E.G.P. was supported by ANR-10-LABX-73 and 2016-StG-715320.

## Author contributions

C.Schu., T.W., and R.T. designed the study, performed experiments and wrote the paper. D.E., D.M., G.P., and C.Schl. performed key experiments and discussed data. L.L., V.S., R.J.V., J.S., K.B., K.K., and L.K. carried out experiments. C.Str., M.F., and S.M. discussed data and revised the manuscript. H.CI.-A. and A.T. carried out confocal microscopy imaging. J.D.M., Y.K., S.E., and S.M. provided mice or important laboratory tools. C.W. and E.G.P. provided key tools, discussed data and revised the manuscript.

## Funding

## Competing interests

The authors declare no competing interests.
