## [Peer Review File · Nature Communications]

Editorial Note: Parts of this peer review file have been redacted as indicated to remove third-party material where no permission to publish could be obtained.

Reviewers' comments:

Reviewer #1 (Remarks to the Author):

This manuscript is mainly using genetic methods to label and microscopy to trace cell lineages, as well as flow cytometry methods for population analysis. All these experiments were designed and performed with highly satisfactory quality.

This solid study describes the presence of unique YS-derived macrophage population. However, it has not revealed for the fundamental intrinsic differences between the YS vs BM populations, given that the YS population exists independently of the monocytic precursors and possesses an extraembryonic origin. More mechanistic insights could be achieved by further characterization of the YS-derived macrophages or ideally and if technically feasible, a single-cell transcriptional profiling study. This may shed more light on their fundamental differences, but I consider these avenues to be beyond the scope of this study.

Another issue is that since *Csf1-Csf1r* is a well-known master and canonic axis in macrophage growth and differentiation, and it's not surprising to see its deletion depletes macrophages and perturbs their homeostasis in the arterial wall. The question is why the BM population displays an enhanced response, and that remains open. This probably warrants a bit more discussion.

A few minor issues:

1. Supplemental Table 1: the dilution and catalog numbers for the antibodies should be added.
2. Row 49 Analogon should be analogue or analog
3. Row 106 *Csf1rMeriCreMer* should be *Csf1rMerCreMer*
4. Row 185 standard abbreviation EdU instead of EDU

Signed: Richard T. Lee

Reviewer #2 (Remarks to the Author):

The paper by Weinberger et al is a very detailed and largely descriptive paper on the origin, homeostasis and reaction to inflammatory stimuli of aortic macrophages. The paper combines many lineage tracing models to reliably determine and study the origin of aortic macrophages. Experiments are well performed.

Major concerns:

1. I feel that the authors can strengthen the manuscripts by characterising aortic macrophages of YS and BM origin in the conditions described (ageing, homeostasis, inflammation). This can be done by sorting, (sc)RNAseq, or CYTOF of the YS vs BM derived macrophages. Do they also have a different secretome?
2. Bearing this in mind, it is important to know whether macrophage subsets develop within the aortic YS or BM derived macrophage populations.
3. It is not clear what the location of the YS vs BM derived macrophages in the aorta is. In the *CSFr1MCMROSAYFP* mouse,, YS macrophages are in the adventitia, whereas they are randomly distributed in other models. How can this be explained??
4. The ANGII model is not the best model to reflect aortic diseases, other models, i.e. atherosclerosis would be better to study.

Reviewer #3 (Remarks to the Author):

The manuscript by Weinberger et. al. investigates the origin, maintenance, and function of arterial resident (adventitia) macrophages. Using lineage-tracing mouse models, they describe dual-origins of arterial macrophages, from yolk-sac (YS) and HSC-origins. Further, they describe a maintenance program within the artery during homeostasis through cell proliferation and dependence on Csf1-signaling. Finally, the authors test the relevance of macrophage origin in the AngII model, showing that macrophage expansion within the artery wall is dependent upon recruitment of progenitors from circulation for BM-derived macrophages and proliferation for YS-derived macrophages. This is an interesting finding.

Overall, the data in the manuscript support the central conclusions being argued, but additional supportive data would dramatically strength the clarity of the paper. Models used in the study are carefully designed with appropriate approaches to address the questions posed. Understanding the role of arterial macrophages to cardiovascular models, such as AngII, is relevant to the field. However, the primary conclusions of the study are merely supportive of those previously described in the literature and do little to extend beyond the prior work. Inclusion of expanded mechanistic insight of the cell types of interest will strengthen and clarify the novel discoveries drawn by the study. In addition, inclusion of missing control and discussion of important published literature will help establish the significance of this paper in the field.

1. The authors state in the abstract and multiple times in the manuscript that “quantitative contribution of YS and BM hematopoiesis to macrophage populations in the aorta of adult mice has been unclear”, which disregards the published literature from the Robbins group establishing dual-origins of arterial macrophages in the adult (including quantitation). Figure 2 from [Ensan et. al. Nature Immunology 2016, PMID #26642357] describes YS contributions in post-natal mice from 0-80 days, and also shows Flt3cre-labeling experiments at the day of birth and adult aorta to account for remaining non-YS derived cells. Despite the authors citing this paper, their contributions were not referenced appropriately in multiple citations. Conclusions and experiments drawn from this prior work needs to be clearly stated in the manuscript and an explanation provided for any different outcomes in experiments discussed.
2. There are only limited descriptions of where aorta samples are being imaged. Based on the layers of elastic lamina, it appears to be on the descending thoracic aorta. Could the authors also comment on whether their observations were consistent in other regions of that aorta (such as abdominal or locations such as the carotid or femoral artery)?
3. For fate-mapping studies (Figure 1A-D, and elsewhere), please include additional control data- The standard in the field has typically been to include brain microglia as a positive control for maximal labeling, then also include blood leukocyte/monocyte labeling at the time of sacrifice as an additional negative control for potential “background” labeling. Inclusion of this data will improve clarity of the findings for the reader.
4. Rankcre data is not interpretable as presented- the authors should show labeling efficiency in their hands and not merely cite a previous article (blood and microglia labeling controls). However, the data is of significant interest. Do the authors observe elevated levels of YS-derived macrophages in the aorta at birth using this model? The Robbins group previously reported 60% labeling at day of birth with a rapid loss as the mouse aged, with the cells being replaced over time by Flt3-labeled cells. Expanding these experiments by performing a time course experiment would greatly improve the manuscript and strengthen conclusions.

5. Csf1r signaling has previously been shown to regulate arterial macrophage maintenance through the use of pharmacologic inhibitor (Ki20227) and also genetic mouse model (Lyve1-cre x Csf1Rfl/fl). This work should be cited and discussed thoroughly [Lim, et. al. Immunity 2018, PMID #30054204].

6. The authors work to differentiate the response of macrophages from different origin on proliferation in response to aging and AngII is of interest to the field and should be thoroughly discussed and expanded. Does proliferation occur in specific regions? Considering the cells in close localization to SMA+ cells as described in the Lim Immunity paper? Could published scRNAseq data on arterial macrophages (Cochain Cir Res 2018) help to inform their different responses to injury?

Minor Comments:

A- Please include detailed description of how many times independent experiments were performed and how many animals were included in each experiment. It is only mentioned sporadically in the manuscript.

B- Please describe the cKit experiment (Figure 3A-B) in more detail including additional controls. If the authors want to draw a conclusion that BM-derived cells are slowly seeding the artery macrophage population, they should perform a time-course experiment to show that it expands with time. In a single cross-section of time, it is not clear if a small percentage of the cells are recombining the fluorescent molecule in the artery or whether cells are coming from new progenitors.

C- What is the rationale for providing Edu every other day (day 5, 3, 1)? Prior experiments performed daily injections, and could the investigators please justify why they conclude a low-level of proliferation (Figure 2E)? Did they compare against other macrophage populations?

D- Please place labels for the gating % in the flow plots (Figure 4)

E- Please include replicate data for Figure 5B w/ stats.

F- Line 312-313: says GFP when I think it should be YFP.

G- Figure 6F: please show blood or BM labeling data.

H- Citations of scRNAseq data of murine aorta leukocytes should include the [Winkels et. al. Circulation Research 2018, PMID #29545366] citation as well since they were published back-to-back.

I- The Supplemental Figure legends need proofing, multiple typos and more detailed descriptions would be welcomed.

Reviewer #1:

1. This manuscript is mainly using genetic methods to label and microscopy to trace cell lineages, as well as flow cytometry methods for population analysis. All these experiments were designed and performed with highly satisfactory quality.

We thank the reviewer for this positive feedback regarding the quality of our analysis.

2. This solid study describes the presence of unique YS-derived macrophage population. However, it has not revealed for the fundamental intrinsic differences between the YS vs BM populations, given that the YS population exists independently of the monocytic precursors and possesses an extraembryonic origin. More mechanistic insights could be achieved by further characterization of the YS-derived macrophages or ideally and if technically feasible, a single-cell transcriptional profiling study. This may shed more light on their fundamental differences, but I consider these avenues to be beyond the scope of this study.

As suggested by the reviewer, we carried out scRNAseq in both steady state conditions and in response to AngII inflammation. We labelled YS-derived macrophages with YFP using *Rank^{Cre}Rosa26^{eYFP}* mice, and then mapped YFP expression in the transcriptional profiling. Excitingly, YFP-derived associated mainly with a specific cluster of adventitial macrophages in steady state and also in inflammatory conditions, in which adventitial macrophages were more heterogenous. The respective cluster of YFP-expressing macrophages was enriched in genes associated with homeostatic and anti-inflammatory properties. Further, we identified cluster-specific expression of genes linked to biological functions, such as *Lyve-1*, which controls arterial collagen expression thereby regulating vascular stiffness and blood pressure functions ¹. In additional bioinformatics analysis, we applied the ClueGo algorithm, which integrates Gene Ontology (GO) terms and KEGG/BioCarta pathways allowing biological interpretation of our transcriptional profiling. These datasets provided distinct functional annotation of biological networks associated with YFP+ and YFP- macrophage populations. In summary, we established large novel datasets providing in-depth information on fundamental differences of macrophage populations of distinct origins in steady state and in response to AngII inflammation.

3. Another issue is that since *Csf1-Csf1r* is a well-known master and canonic axis in macrophage growth and differentiation, and it's not surprising to see its deletion depletes macrophages and perturbs their homeostasis in the arterial wall. The question is why the BM population displays an enhanced response, and that remains open. This probably warrants a bit more discussion.

We thank the reviewer for this comment. We expanded the discussion section as follows: “The chemokine *CSF1* provides essential homeostatic signaling for the growth and differentiation of macrophages ^{2,3}. To evaluate the contribution of the BM to the recovery of the arterial macrophage pool after cessation of *CSF1R*-inhibition, a BM-chimera model was implemented. In line with other

studies using irradiation dependent chimera models ⁴, they found substantial replacement of resident by BM-derived macrophages already in steady state and near-complete replacement after recovery from CSF1R-mediated macrophage depletion. To circumvent the bias of irradiation and its described consequences, we used the *Flt3*-reporter mice to determine the contribution of BM-derived macrophages to macrophage recovery. When we interfered with CSF1R signalling, BM-derived arterial macrophages displayed a stronger ablative response compared to their YS-derived counterparts. A similar effect of CSF1R-inhibition has been reported in the lung, where BM-derived interstitial macrophages show increased depletion compared to embryo-derived alveolar macrophages ⁵. After acute ablation, arterial macrophages recovered quickly and numbers returned to baseline within 7 days. We found that the recovery of the tissue macrophage pool was driven by local proliferation of embryonic and also BM-derived cells, when proliferation capacity of arterial macrophages was not affected by radiation. Taken together, we provide evidence that survival and proliferation of both BM- and YS- derived macrophages is driven by CSF1R signalling, identifying it as an essential factor of macrophage homeostasis in the arterial wall.”

4. A few minor issues:

(1) Supplemental Table 1: the dilution and catalog numbers for the antibodies should be added.

(2) Row 49 Analogon should be analogue or analog

(3) Row 106 *Csf1rMeriCreMer* should be *Csf1rMerCreMer*

(4) Row 185 standard abbreviation EdU instead of EDU

We made all changes as suggested.

Reviewer #2:

1. The paper by Weinberger et al is a very detailed and largely descriptive paper on the origin, homeostasis and reaction to inflammatory stimuli of aortic macrophages. The paper combines many lineage tracing models to reliably determine and study the origin of aortic macrophages. Experiments are well performed.

We thank the reviewer for this comment. We revised the article and now provide more details

2. I feel that the authors can strengthen the manuscripts by characterising aortic macrophages of YS and BM origin in the conditions described (ageing, homeostasis, inflammation). This can be done by sorting, (sc)RNAseq, or CYTOF of the YS vs BM derived macrophages. Do they also have a different secretive?

As suggested by the reviewer, we carried out scRNAseq at both steady state and in response to AngII inflammation. We labelled YS-derived macrophages with YFP using *Rank^{Cre}Rosa26^{eYFP}* mice, and then mapped YFP expression in the transcriptional profiling. Excitingly, YFP-derived associated mainly with a specific cluster of adventitial macrophages in steady state and also in inflammatory conditions, in which adventitial macrophages were more heterogenous. The respective cluster of YFP-expressing macrophages was enriched in genes associated with homeostatic and anti-inflammatory properties. We also identified cluster-specific expression of genes linked to biological functions, such as *Lyve-1*, which controls arterial collagen expression thereby regulating vascular stiffness and blood pressure functions ¹. We carried out additional bioinformatics analysis applying the ClueGo algorithm, which integrates Gene Ontology (GO) terms and KEGG/BioCarta pathways allowing biological interpretation of our transcriptional profiling. These datasets provided distinct functional annotation of biological networks, including secretive functions, associated with YFP+ and YFP- macrophage populations. In summary, by establishing these large novel datasets we provide in-depth information on fundamental differences of macrophage populations of distinct origins in steady state and in response to AngII inflammation.

3. Bearing this in mind, it is important to know whether macrophage subsets develop within the aortic YS or BM derived macrophage populations.

We thank the reviewer for this comment. Indeed, by establishing transcriptomic profiling at single cell resolution, we found that adventitial macrophages were heterogenous.

YS-derived macrophages associated mainly with cluster 0 in steady state, and with cluster 2 in AngII inflammation. Their minor association with other clusters could indicate the presence of subsets.

BM-derived macrophages displayed greater heterogeneity in both steady state and in response to AngII inflammation. In AngII inflammation, we identified 5 macrophage clusters, which is comparable to studies in mouse atherosclerosis ⁶⁻⁸.

4. It is not clear what the location of the YS vs BM derived macrophages in the aorta is. In the CSFr1MCMROSA^{YFP} mouse, YS macrophages are in the adventitia, whereas they are randomly distributed in other models. How can this be explained?

To further address the distribution of macrophage populations in the mouse aorta, we carried out additional analysis of *Flt3^{Cre}Rosa26^{eYFP}* mice. We quantified YFP+ and YFP- macrophages in 5 different regions of the aorta (from the aortic arch to the distal part of the descending aorta), counting more than 10.000 macrophages on histological sections. We determined that the distribution of YFP+ macrophages is similar between the different anatomic regions. This information was included in supplemental figure 4.

In light of these new insights, we excluded the phrase “cluster of macrophages” describing the distribution of YS-derived macrophages in pulse labeling experiments using the *Csf1r^{MCM}Rosa26^{eYFP}* mice. The observed clustering in the pulse-labelling experiments may be explained by the low labelling efficiency which in itself leads to a certain clustering of cells.

5. The ANGII model is not the best model to reflect aortic diseases, other models, i.e. atherosclerosis would be better to study.

We agree that atherosclerosis is of pivotal interest. We now plan to continue our research in this direction and aim to address YS macrophages in atherosclerotic lesion development and regression.

Reviewer #3:

1. The manuscript by Weinberger et. al. investigates the origin, maintenance, and function of arterial resident (adventitia) macrophages. Using lineage-tracing mouse models, they describe dual-origins of arterial macrophages, from yolk-sac (YS) and HSC-origins. Further, they describe a maintenance program within the artery during homeostasis through cell proliferation and dependence on Csf1-signaling. Finally, the authors test the relevance of macrophage origin in the AngII model, showing that macrophage expansion within the artery wall is dependent upon recruitment of progenitors from circulation for BM-derived macrophages and proliferation for YS-derived macrophages. This is an interesting finding.

Overall, the data in the manuscript support the central conclusions being argued, but additional supportive data would dramatically strength the clarity of the paper. Models used in the study are carefully designed with appropriate approaches to address the questions posed. Understanding the role of arterial macrophages to cardiovascular models, such as AngII, is relevant to the field. However, the primary conclusions of the study are merely supportive of those previously described in the literature and do little to extend beyond the prior work. Inclusion of expanded mechanistic insight of the cell types of interest will strengthen and clarify the novel discoveries drawn by the study. In addition, inclusion of missing control and discussion of important published literature will help establish the significance of this paper in the field.

We thank the reviewer for this evaluation. In the revision process, we generated a large body of new data. This includes scRNAseq of adventitial immune cells mapping macrophage ontogeny in both steady state and AngII inflammation. Expression of cluster-specific genes such as *Lyve-1*, provided mechanistic insight into different function of YS and BM macrophages. We carried out additional bioinformatics analysis to interrogate biological networks associated with gene expression. We believe that with establishing these large novel datasets, our study extends well beyond prior work. In addition, we included additional information on the labeling of blood monocytes and brain microglia in respective mouse models. Further, we thoroughly revised the manuscript text and amended the citations of previous reports to better frame the current findings of our fate mapping mice as well as the AngII inflammation model.

2. The authors state in the abstract and multiple times in the manuscript that “quantitative contribution of YS and BM hematopoiesis to macrophage populations in the aorta of adult mice has been unclear”, which disregards the published literature from the Robbins group establishing dual-origins of arterial macrophages in the adult (including quantitation). Figure 2 from [Ensan et. al. Nature Immunology 2016, PMID #26642357] describes YS contributions in post-natal mice from 0-80 days, and also shows *Flt3cre*-labeling experiments at the day of birth and adult aorta to account for remaining non-YS derived cells. Despite the authors citing this paper, their contributions were not referenced appropriately in multiple citations. Conclusions and experiments drawn from this prior work needs to be clearly stated in the manuscript and an explanation provided for any different outcomes in experiments discussed.

We apologize for inappropriate referencing and incomplete discussion of the papers by Ensan and Robbins. We thoroughly revised the manuscript, amended citations and changed the discussion. We hope that the reviewer considers the changes made as appropriate.

To provide another new set of data during manuscript revision, we established a time-course analysis of both absolute and relative (% YFP expression) numbers of EMP-derived macrophages in the adventitia of *Rank^{Cre}Rosa26^{eYFP}* fate mapping mice. We confirmed that arterial macrophages were mostly derived from YS EMPs in the first week of life (Fig. 1E-H), which had been shown previously ⁴. From birth to adulthood, the absolute number of macrophages in the adventitia increased approximately 3.7 fold (Fig. 1G). This was driven by an increase of mainly EMP-derived macrophages in adolescence as well as through contribution of BM-derived (eYFP-) macrophages later in adulthood. While this resulted in a relative decline of eYFP+ macrophages, they remained the dominant macrophage population in mice until at least 45 weeks of age. In the process of ageing, EMP-derived macrophages diminished in numbers. However, their loss in 90 weeks-old mice was not compensated for by BM-derived (eYFP-) macrophages, whose population remained stable in absolute numbers (Fig. 1G). Thus, EMP-derived macrophages are the predominant macrophage population in healthy adult mice but are diminished in the course of ageing. We believe that this novel dataset extends previous publications and will be of interest to the reader.

3. There are only limited descriptions of where aorta samples are being imaged. Based on the layers of elastic lamina, it appears to be on the descending thoracic aorta. Could the authors also comment on whether their observations were consistent in other regions of that aorta (such as abdominal or locations such as the carotid or femoral artery)?

We thank the reviewer for this important comment.

For flow cytometry, we analyzed the aorta between the left subclavian artery and the aortic bifurcation. We now provide this information more clearly in the manuscript.

To further address the distribution of macrophage populations in the mouse aorta, we carried out additional analysis of *Flt3^{Cre}Rosa26^{eYFP}* mice. We quantified YFP+ and YFP- macrophages in 5 different regions of the aorta (from the aortic arch to the distal part of the descending aorta), counting more than 10,000 macrophages on histological sections. We determined that the distribution of YFP+ macrophages is similar between the different anatomic regions. This information was included in supplemental figure 4.

4. For fate-mapping studies (Figure 1A-D, and elsewhere), please include additional control data- The standard in the field has typically been to include brain microglia as a positive control for maximal labeling, then also include blood leukocyte/monocyte labeling at the time of sacrifice as an additional negative control for potential “background” labeling. Inclusion of this data will improve clarity of the findings for the reader.

We agree with the reviewer on the usage of brain microglia as well as blood monocytes as reference points. We now include this information in various experiments, i.e. *Rank^{Cre}Rosa26^{eYFP}* mice (Fig. 1F), *Flt3^{Cre}Rosa26^{eYFP}* mice (Fig. 3B), as well as the *Mx1^{Cre}Myb^{fllox}* chimera model (Fig. 3H, Fig. 5F).

5. Rankcre data is not interpretable as presented- the authors should show labeling efficiency in their hands and not merely cite a previous article (blood and microglia labeling controls). However, the data is of significant interest. Do the authors observe elevated levels of YS-derived macrophages in the aorta at birth using this model? The Robbins group previously reported 60% labeling at day of birth with a rapid loss as the mouse aged, with the cells being replaced over time by Flt3-labeled cells. Expanding these experiments by performing a time course experiment would greatly improve the manuscript and strengthen conclusions.

We thank the reviewer for this comment. As discussed above in response to comments 2 and 4, we now provide information on labeling efficiency of blood and microglia. We show that the relative number of YS-derived (YFP+) macrophages is high at postnatal day 3 (Fig. 1F, G). We also quantified total numbers of macrophages in the adventitia and established a time course analysis up to 90 weeks. We agree with the reviewer that these experiments greatly improved the manuscript.

6. Csf1r signaling has previously been shown to regulate arterial macrophage maintenance through the use of pharmacologic inhibitor (Ki20227) and also genetic mouse model (Lyve1-cre x Csf1Rfl/fl). This work should be cited and discussed thoroughly [Lim, et. al. Immunity 2018, PMID #30054204].

We thank the reviewer for this comment. We now cite this important paper in our revised manuscript in various paragraphs. The respective paper by Lim et al. also provided pivotal information on the role of LYVE1+ macrophages in arterial biology. Since in the scRNAseq analysis we identified *Lyve-1* as a signature gene of cluster 0 (steady state conditions), in which YS-derived macrophages were enriched, the findings strengthen the notion that YS and BM derived macrophages are associated with different functions.

Use of the pharmacologic inhibitor Ki20227 is now cited in the results section: "Survival of resident arterial macrophages is dependent on CSF1R as shown in a genetic model targeting Lyve-1-expressing macrophages as well as using the pharmacological CSF1R-inhibitor Ki20227 ³."

To address the importance of Csf1r signaling in more detail, we expanded the discussion section as follows: "*The chemokine CSF1 provides essential homeostatic signaling for the growth and differentiation of macrophages* ^{2,3}. To evaluate the contribution of the BM to the recovery of the arterial macrophage pool after cessation of CSF1R-inhibition, a BM-chimera model was implemented. In line with other studies using irradiation dependent chimera models ⁴, they found substantial replacement of resident by BM-derived macrophages already in steady state and near-

complete replacement after recovery from CSF1R-mediated macrophage depletion. To circumvent the bias of irradiation and its described consequences, we used the *Flt3*-reporter mice to determine the contribution of BM-derived macrophages to macrophage recovery. When we interfered with CSF1R signalling, BM-derived arterial macrophages displayed a stronger ablative response compared to their YS-derived counterparts. A similar effect of CSF1R-inhibition has been reported in the lung, where BM-derived interstitial macrophages show increased depletion compared to embryo-derived alveolar macrophages⁵. After acute ablation, arterial macrophages recovered quickly and numbers returned to baseline within 7 days. We found that the recovery of the tissue macrophage pool was driven by local proliferation of embryonic and also BM-derived cells, when proliferation capacity of arterial macrophages was not affected by radiation. Taken together, we provide evidence that survival and proliferation of both BM- and YS- derived macrophages is driven by CSF1R signalling, identifying it as an essential factor of macrophage homeostasis in the arterial wall.”

7. The authors work to differentiate the response of macrophages from different origin on proliferation in response to aging and AngII is of interest to the field and should be thoroughly discussed and expanded. Does proliferation occur in specific regions? Considering the cells in close localization to SMA+ cells as described in the Lim Immunity paper? Could published scRNAseq data on arterial macrophages (Cochain Cir Res 2018) help to inform their different responses to injury?

We thank the reviewer for this question. Considering the paper by Lim et al., the cells in close localization to SMA+ cells were characterized by expression of *Lyve-1*¹. We identified *Lyve-1* in our transcription analysis as a top upregulated gene in the cluster of YS-derived macrophages. In the paper by Lim et al., the LYVE1+ macrophage population controlled arterial collagen expression thereby regulating vascular stiffness and blood pressure functions. This provides an important functional aspect.

scRNAseq data on arterial macrophages in AngII inflammation has not been reported to date. To determine differences in macrophage responses to AngII injury, we carried out *de novo* scRNA-seq analysis in *Rank^{Cre}Rosa26^{eYFP}* fate mapping mice, in which tissue macrophages of early embryonic origin were labelled with YFP. We now provide data on the responses of adventitial immune cells to AngII inflammation in single cell resolution (new figures 6 and 7, and supplemental data). We thereby identified specific responses of YFP+ macrophages. Referring to the reviewer's question, we carried out additional bioinformatics analysis applying the ClueGo algorithm, which integrates Gene Ontology (GO) terms and KEGG/BioCarta pathways allowing biological interpretation of our transcriptional profiling. These datasets provided distinct functional annotation of biological networks associated with YFP+ and YFP- macrophage populations.

We expanded the analysis of YFP+ macrophages of *Rank^{Cre}Rosa26^{eYFP}* fate mapping mice, and quantified macrophage numbers in the adventitia in a time course analysis. We show that YFP+ macrophages increase in numbers in postnatal life and are lost in ageing (90 weeks-old) mice. This provides a novel aspect on the proliferation and homeostasis of these macrophages. However, further studies are needed to address the biology of different macrophage populations in later life.

Finally, we expanded and revised the discussion on macrophage proliferation in the discussion section.

8. Minor Comments: Please include detailed description of how many times independent experiments were performed and how many animals were included in each experiment. It is only mentioned sporadically in the manuscript.

We now provide this information in the figures legends. We also provide source data for all experiments.

9. Please describe the cKit experiment (Figure 3A-B) in more detail including additional controls. If the authors want to draw a conclusion that BM-derived cells are slowly seeding the artery macrophage population, they should perform a time-course experiment to show that it expands with time. In a single cross-section of time, it is not clear if a small percentage of the cells are recombining the fluorescent molecule in the artery or whether cells are coming from new progenitors.

We fully agree with the reviewer on the interpretation of the cKit experiment. Unfortunately, respective mice were not available to us during the Covid-19 pandemic and additional experiments could not be carried out. We then shifted the focus on the scRNA-seq analysis and made every effort to successfully complete those experiments.

In the revised manuscript, we therefore toned down the interpretation of the cKit experiment. We would like to keep the data at the end of figure 3 as an add-on experiment in addition to the characterization of *Flt3^{Cre}Rosa26^{eYFP}* lineage tracing mice and the *Mx1^{Cre}Myb^{fllox}* bone marrow chimeric mice.

10. What is the rationale for providing Edu every other day (day 5, 3, 1)? Prior experiments performed daily injections, and could the investigators please justify why they conclude a low-level of proliferation (Figure 2E)? Did they compare against other macrophage populations?

The EdU protocol was based on the publications by Ensan et al.⁴ and Nawaz et al.⁹, who applied the compound 3-4 times every other day, respectively. The interpretation of low-level proliferation was based on the comparison of steady state levels to proliferation in AngII inflammation and following PLX depletion. In the revised manuscript, we toned down this interpretation and rephrased the respective paragraph.

We did not specifically compare against other macrophage populations in our experiments. Indeed, there is some evidence on the proliferative capacity of other tissue macrophage populations such as for example Langerhans cells in the epidermis. Notably, the proliferative capacity in steady state as well as their expansion in postnatal life is comparable to that of adventitial macrophages. We now include this interesting aspect in the discussion section, which

reads as follows: “Local proliferation in tissues is an important macrophage feature, allowing them to maintain their populations but also to adapt to changing conditions ¹⁰. In the first week of life, Langerhans cells in the epidermis undergo a boost of local proliferation ¹¹. Later on, these macrophages are maintained independently of BM HSCs ¹².”

11. Please place labels for the gating % in the flow plots (Figure 4)

In this respective figure, we analyzed whole adventitia from individual mice and display total number of macrophages. Thus, we believe that the depletion of macrophages is readily visible to the reader and that indicating relative % would rather be confusing. We hope that this is acceptable for the reviewer. We adapted the figure legend and denote more clearly that absolute numbers of macrophages are displayed.

12. Please include replicate data for Figure 5B w/ stats.

We now provide this information in the figures legends. We also provide source data for all experiments.

13. Line 312-313: says GFP when I think it should be YFP.

We made respective change.

14. Figure 6F: please show blood or BM labeling data.

We now provide this data.

15. Citations of scRNAseq data of murine aorta leukocytes should include the [Winkels et. al. Circulation Research 2018, PMID #29545366] citation as well since they were published back-to-back.

We thank the reviewer for this comment. We included the respective landmark paper.

16. The Supplemental Figure legends need proofing, multiple typos and more detailed descriptions would be welcomed.

We revised all supplemental figure legends.

References

1. Lim HY, Lim SY, Tan CK, et al. Hyaluronan Receptor LYVE-1-Expressing Macrophages Maintain Arterial Tone through Hyaluronan-Mediated Regulation of Smooth Muscle Cell Collagen. *Immunity* 2018;49:326-341.e327.
2. Ginhoux F, Greter M, Leboeuf M, et al. Fate mapping analysis reveals that adult microglia derive from primitive macrophages. *Science (New York, NY)* 2010;330:841-845.
3. Lim HY, Lim SY, Tan CK, et al. Hyaluronan Receptor LYVE-1-Expressing Macrophages Maintain Arterial Tone through Hyaluronan-Mediated Regulation of Smooth Muscle Cell Collagen. *Immunity* 2018;49:1191.
4. Ensan S, Li A, Besla R, et al. Self-renewing resident arterial macrophages arise from embryonic CX3CR1(+) precursors and circulating monocytes immediately after birth. *Nature immunology* 2016;17:159-168.
5. Tan SY, Krasnow MA. Developmental origin of lung macrophage diversity. *Development* 2016;143:1318-1327.
6. Lin JD, Nishi H, Poles J, et al. Single-cell analysis of fate-mapped macrophages reveals heterogeneity, including stem-like properties, during atherosclerosis progression and regression. *JCI insight* 2019;4.
7. Cochain C, Vafadarnejad E, Arampatzi P, et al. Single-Cell RNA-Seq Reveals the Transcriptional Landscape and Heterogeneity of Aortic Macrophages in Murine Atherosclerosis. *Circulation research* 2018;122:1661-1674.
8. Winkels H, Ehinger E, Vassallo M, et al. Atlas of the Immune Cell Repertoire in Mouse Atherosclerosis Defined by Single-Cell RNA-Sequencing and Mass Cytometry. *Circulation research* 2018;122:1675-1688.
9. Nawaz A, Aminuddin A, Kado T, et al. CD206(+) M2-like macrophages regulate systemic glucose metabolism by inhibiting proliferation of adipocyte progenitors. *Nature communications* 2017;8:286.
10. Perdiguero EG, Geissmann F. The development and maintenance of resident macrophages. *Nature immunology* 2016;17:2-8.
11. Chorro L, Sarde A, Li M, et al. Langerhans cell (LC) proliferation mediates neonatal development, homeostasis, and inflammation-associated expansion of the epidermal LC network. *The Journal of experimental medicine* 2009;206:3089-3100.
12. Kanitakis J, Morelon E, Petruzzo P, Badet L, Dubernard JM. Self-renewal capacity of human epidermal Langerhans cells: observations made on a composite tissue allograft. *Experimental dermatology* 2011;20:145-146.

REVIEWER COMMENTS

Reviewer #1 (Remarks to the Author):

NA

Reviewer #2 (Remarks to the Author):

The authors have answered all the reviewer's questions, and performed the required additional experiments. No additional comments.

This is a very nice paper!

Reviewer #3 (Remarks to the Author):

In the revised manuscript by Weinberger et. al. the authors address the contribution of ontogeny in arterial macrophage homeostasis and in a model of vascular inflammation. I appreciate the inclusion of requested controls and adjustments to conclusions from the previous submission. Addition of new single cell sequencing data dramatically strengthen the novelty of the paper and helps the authors draw important conclusions regarding the contribution of embryonic-derived macrophages in homeostasis and disease. Yet, additional analysis of this data would improve interpretation of the role that origin plays in controlling macrophage function.

1. scRNA-seq data described in figure 2 has limited cell numbers included in analysis with high gene coverage; 346 total cells with median 2,748 unique genes/cell. Therefore, it is possible that a full range of cellular diversity and even rare populations present in the steady state are not detected. This concern should be described for the reader, so that data are not over-interpreted. Conversely, the second scRNA-seq dataset represents a large number of cells, but at a comparatively low coverage; 4,419 total cells with median 687 unique genes/cell, which creates an alternative concern.

2. Gene expression analysis comparing YFP+ and YFP- macrophages from Cluster 0 in the steady state should be compared and described. This is a central premise of the paper that would extend beyond prior work in the field and is not sufficiently addressed in Figure 2.

3. The analysis in Figure 7A is not an appropriate approach, as it does not correct for the different contribution of yolk-sac-derived cells within each macrophage subpopulation. Since YFP+ cells are preferentially represented by cluster 2, many of the genes that are subsequently enriched in Fig7 are overlapping with cluster 2 and are likely not shared by other YFP+ macrophages from cluster 0, 3, 4,7. A more appropriate analysis will be to assess gene expression changes within each sub-cluster of macrophages of YFP+ vs. YFP-, then follow with analysis of overlapping signatures that are shared within the YFP+ macrophages between clusters. Lastly, the figure format appears to be misleading by using scaled expression- it gives the appearance that genes are either expressed or not expressed (blue or grey), when in fact many genes are expressed to different degrees. Average gene expression may be a more appropriate way to show this type of graph.

4. If possible, the homeostatic and AngII scRNA-seq datasets should be combined into a third analysis to allow for direct comparison of responses between states and to determine changes within clusters/cell origins in response to injury.

Minor

1. Lines 178, 224, 659: should be corrected to say that total CD45+ cells isolated from aorta, since no enrichment of adventitia-specific cells was performed.
2. Please include description of the 10x chemistry version used for scRNA-seq analysis.

Rebuttal letter to NCOMMS-19-33302A

Title “Ontogeny of arterial macrophages defines their response to homeostatic signaling and angiotensin II-induced inflammation”

We thank the reviewers for their thorough review and positive feedback on our revised manuscript. In response to the comments of reviewer #3, we carried out additional bioinformatic analyses and revised the manuscript accordingly. A detailed point-by-point response to the reviewers' comments is given below.

Reviewer #1 (Remarks to the Author):

NA

Reviewer #2 (Remarks to the Author):

The authors have answered all the reviewer's questions, and performed the required additional experiments. No additional comments. This is a very nice paper!

We thank the author for this very positive comment.

Reviewer #3 (Remarks to the Author):

In the revised manuscript by Weinberger et. al. the authors address the contribution of ontogeny in arterial macrophage homeostasis and in a model of vascular inflammation. I appreciate the inclusion of requested controls and adjustments to conclusions from the previous submission. Addition of new single cell sequencing data dramatically strengthen the novelty of the paper and helps the authors draw important conclusions regarding the contribution of embryonic-derived macrophages in homeostasis and disease. Yet, additional analysis of this data would improve interpretation of the role that origin plays in controlling macrophage function.

We thank the reviewer for the positive feedback on our revised manuscript and for pointing out its novelty.

1. scRNA-seq data described in figure 2 has limited cell numbers included in analysis with high gene coverage; 346 total cells with median 2,748 unique genes/cell. Therefore, it is possible that a full range of cellular diversity and even rare populations present in the steady state are not detected. This concern should be described for the reader, so that data are not over-interpreted. Conversely, the second scRNA-seq dataset represents a large number of cells, but at a comparatively low coverage; 4,419 total cells with median 687 unique genes/cell, which creates an alternative concern.

We thank the reviewer for this comment. We agree with the limitations raised and have incorporated their description in the revised text. We would like to point out that the total number of cells isolated at steady state is in full agreement with published articles ^{1, 2}. Nonetheless, it poses a limitation and we now highlight this aspect in the results section. The revised text reads as follows: “A limitation of our scRNA-seq analysis in steady state is the low number of total cells analyzed. This is due to the low abundance of macrophages in healthy adventitia, which has also

been observed in other reports ^{1,2}.” In the results section on AngII inflammation, we added the following sentence: “It should be noted that a potential limitation of our scRNA-seq analysis in AngII conditions is the relatively low coverage.”

2. Gene expression analysis comparing YFP+ and YFP- macrophages from Cluster 0 in the steady state should be compared and described. This is a central premise of the paper that would extend beyond prior work in the field and is not sufficiently addressed in Figure 2.

We thank the reviewer for this comment. We carried out additional gene expression analysis comparing eYFP+ and eYFP- macrophages from Cluster 0 in steady state. Interestingly, some genes were differentially expressed between macrophage lineages within this cluster. For example *Lyve-1* displayed increased expression in EMP-derived (eYFP+) macrophages. This further supports the notion that ontogeny of macrophages is associated with differences in their cellular identity in adult mice.

The new analysis is now included in Supplemental Figure 4.

3. The analysis in Figure 7A is not an appropriate approach, as it does not correct for the different contribution of yolk-sac-derived cells within each macrophage subpopulation. Since YFP+ cells are preferentially represented by cluster 2, many of the genes that are subsequently enriched in Fig7 are overlapping with cluster 2 and are likely not shared by other YFP+ macrophages from cluster 0, 3, 4, 7. A more appropriate analysis will be to assess gene expression changes within each sub-cluster of macrophages of YFP+ vs. YFP-, then follow with analysis of overlapping signatures that are shared within the YFP+ macrophages between clusters.

We thank the reviewer for this comment. We carried out additional comparisons of eYFP+ and eYFP- macrophages of each cluster in AngII inflammation. Indeed, eYFP+ and eYFP- macrophages of the same cluster exhibited differences in their gene expression. While differences were rather low in cluster 2 (“homeostatic” macrophages), they were more prominent in others

such as cluster 7 (“inflammatory” macrophages). The presence of distinct gene signatures within clusters supports the notion that ontogeny impacts on macrophage gene expression and potentially function.

The new analysis is now included in Supplemental Figure 10.

We also addressed overlapping gene expression of eYFP+ or eYFP- macrophages. We found that some genes were shared between macrophages of different clusters, and we now provide a detailed table of respective genes (Supplemental Figure 11). For example, in the lineage of EMP-derived macrophages, we identified *Cx3cr1* to overlap between clusters 0, 2, and 7.

Interestingly, one gene was shared between all clusters of eYFP+ macrophages in AngII inflammation, which was identified as chemokine (C-C motif) ligand 12 (Ccl12, MCP-5). MCP-5 has previously been identified in activated macrophages³, however, its functional role is incompletely understood and could represent an interesting molecule to address in future studies. The new information and additional analysis is now included Supplemental Figure 11. Again, we would like to thank the reviewer for indicating this analysis.

Lastly, the figure format appears to be misleading by using scaled expression- it gives the appearance that genes are either expressed or not expressed (blue or grey), when in fact many genes are expressed to different degrees. Average gene expression may be a more appropriate way to show this type of graph.

We thank the reviewer for this comment. We now provide a detailed scale that differentiates more clearly the different levels of gene expression. We revised Figure 7A as follows:

4. If possible, the homeostatic and AngII scRNA-seq datasets should be combined into a third analysis to allow for direct comparison of responses between states and to determine changes within clusters/cell origins in response to injury.

The suggested analysis could be interesting but is a huge undertaking. As the reviewer pointed out, the low number of immune cells in steady state arteries represents a potential limitation to the scRNA-Seq analysis. In order to carry out the suggested combined analysis, we first plan in future work to establish additional scRNA-Seq datasets from steady state conditions to increase leukocyte numbers. This would then allow us to carry out a direct comparison between states, and potentially other inflammatory conditions such as atherosclerosis. However, we would consider this future work.

Minor

- 1. Lines 178, 224, 659: should be corrected to say that total CD45+ cells isolated from aorta, since no enrichment of adventitia-specific cells was performed.**

The lines were correct as written. As depicted in the graphical presentation in Figure 2A, in the figure legends (e.g. lines 224 and 510) as well as the main text (e.g. line 178), we carried out single cell analysis of adventitial immune cells.

The adventitia is the major immune cell reservoir of the arterial wall. Leukocytes are found almost exclusively in the adventitia in steady state, as indicated by our extensive histological analysis. In contrast to models of atherosclerosis, which induces strong leukocyte recruitment in intima/media and formation of atherosclerotic lesions in close proximity to the lumen, AngII inflammation increases immune cell numbers mostly in the adventitia. The histological images in Figure 5C illustrate that in response to AngII the number of macrophages increased in the adventitia but remained low in intima/media. We carried out an additional histologic analysis of aorta cross-sections of AngII-treated (10 days) mice, in order to quantify their relative distribution in the different anatomic (intima, media, adventitia) regions. As shown in the graph below, approximately 90% of macrophages are located in the adventitia in AngII inflammation.

Of note, similar findings were made by other groups analyzing macrophage distribution in Ang II inflammation. The figure shown below was published by Rateri et al. ⁴. This respective paper supports the low-level accumulation of macrophages in the aortic intima of AngII-treated mice.

Legend of Supplemental Figure S5 (Rateri et al.): Examples of intimal versus adventitial location of macrophages throughout the length of aneurysmal aortas. Symbols represent macrophage counts in specific areas (intima = yellow, adventitial = red, and total = green) of aortic sections.

—

[Redacted]

We added the quantification on the anatomical location of macrophages in the aorta of AngII-treated mice in the revised supplemental figure 8C. In the results section we added the following sentence: “The AngII-induced increase in macrophages is largely confined to the adventitia, which harbors approximately 90% of macrophages (Suppl. Fig. 8C, Fig. 5C)”.

Since accumulating leukocytes mostly located within the adventitia in this model, we focused the scRNA-Seq analysis on the adventitia. Additional analysis of intima/media leukocytes would have required further processing including an additional step of enzymatic digestion⁵, which would have created a technical bias. However, focusing on the adventitia could also be considered a limitation, despite covering most leukocytes. To acknowledge this aspect, we added the following sentence in the revised results section: “Focusing the scRNA-Seq analysis on the adventitial tissue of the aorta, thereby excluding intima and media, can be considered another limitation. However, we analyzed most leukocytes that accumulated in the aorta in AngII inflammation as demonstrated by extensive histology.”

2. Please include description of the 10x chemistry version used for scRNA-seq analysis.

We now describe the chemistry version used for scRNA-seq in detail in the methods section of the manuscript: “After sorting of adventitial CD45-positive immune cells from whole aortas (aorta ascendens to aortic bifurcation), viable cells were proceeded for single cell capture, barcoding and library preparation using Chromium Next GEM single cell 3’ (v3.1, 10x Genomics) according to manufacturer’s specifications.”

References:

1. Cochain C, Vafadarnejad E, Arampatzi P, et al. Single-Cell RNA-Seq Reveals the Transcriptional Landscape and Heterogeneity of Aortic Macrophages in Murine Atherosclerosis. *Circ Res* 2018;122:1661-1674.
2. Winkels H, Ehinger E, Vassallo M, et al. Atlas of the Immune Cell Repertoire in Mouse Atherosclerosis Defined by Single-Cell RNA-Sequencing and Mass Cytometry. *Circ Res* 2018;122:1675-1688.
3. Sarafi MN, Garcia-Zepeda EA, MacLean JA, Charo IF, Luster AD. Murine monocyte chemoattractant protein (MCP)-5: a novel CC chemokine that is a structural and functional homologue of human MCP-1. *The Journal of experimental medicine* 1997;185:99-109.
4. Rateri DL, Howatt DA, Moorlegghen JJ, Charnigo R, Cassis LA, Daugherty A. Prolonged infusion of angiotensin II in apoE(-/-) mice promotes macrophage recruitment with continued expansion of abdominal aortic aneurysm. *The American journal of pathology* 2011;179:1542-1548.
5. Galkina E, Kadl A, Sanders J, Varughese D, Sarembock IJ, Ley K. Lymphocyte recruitment into the aortic wall before and during development of atherosclerosis is partially L-selectin dependent. *The Journal of experimental medicine* 2006;203:1273-1282.

REVIEWERS' COMMENTS:

Reviewer #3 (Remarks to the Author):

Thank you for addressing all concerns, congratulations on a wonderful paper.